

**Analysis of insoluble particles in hailstones in China**
Haifan Zhang[1], Xiangyu Lin[1], Qinghong Zhang[1*] , Kai Bi[2*], Chan-Pang Ng[1], Yangze Ren[1], Huiwen Xue[1], Li Chen[3], Zhuolin
Chang[4]
[1]Department of Atmospheric and Oceanic Sciences, School of Physics, Peking University, Beijing 100871, China
[2]Field Experiment Base of Cloud and Precipitation Research in North China, China Meteorological Administration, Beijing
101200, China
[3]Electron Microscopy Laboratory, Peking University, Beijing 100871, China
[4]Key Laboratory for Meteorological Disaster Monitoring and Early Warning and Risk Management of Characteristic
Agriculture in Arid Regions, China Meteorological Administration, Yinchuan 750002, China
*_Corresponding author_: Qinghong Zhang (qzhang@pku.edu.cn); Kai Bi (bikai_picard@vip.sina.com)





**Abstract.** Insoluble particles affect weather and climate indirectly by heterogeneous freezing process. Current weather and
climate models have large uncertainty in freezing process simulation due to little regarding species and number concentration
of heterogeneous ice-nucleating particles, mainly insoluble particles. Here, for the first time, size distribution and species of
insoluble particles are analyzed in 30 shells of 12 hailstones in China, using scanning electron microscopy and energy
dispersive X-ray spectrometry. Total 289,461 insoluble particles are detected and grouped into 3 species: organics, dust, and
bioprotein by machine learning methods. The size distribution of insoluble particles of each species vary greatly in different
hailstorms but little in shells. Further, classic size distribution modes of organics and dust were performed as logarithmic
normal distributions, which may be adapted in future weather and climate models though uncertainty still exists. Our finding
suggests that physical properties of aerosols should be considered in model simulation on ice freezing process.
**1 Introduction**
Insoluble particles, acting as main heterogeneous ice-nucleating particles in the atmosphere(Lamb and Verlinde, 2011),
may indirectly impact precipitation formation and radiative forcing(Hoose and Möhler, 2012; DeMott et al., 2015), and further
impact weather and climate(Vergara-Temprado et al., 2018). Temperature and vapor supersaturation are used to calculate the
number concentration of ice crystal particles in microphysics parameterization rather than considering the physical properties
of ice-nucleating particles in weather and climate models(DeMott et al., 2010). Only few models calculate the number
concentration of ice-nucleating particles in clouds, that leads to a misestimation about number concentration of ice particles
and large errors in simulation(Vergara-Temprado et al., 2018).
An improved description for the number concentrations of ice-nucleating particles is needed, while obstructed by a lack
of complete microphysical observation in clouds about ice-nucleating particles(DeMott et al., 2010). Measurements of the
number concentration and species of ice-nucleating particles, mainly insoluble particles(Lamb and Verlinde, 2011), were
conducted by an airborne equipment or laboratory instrument with air parcels, to understand the process of ice nucleation in
clouds(DeMott et al., 2010; Prenni et al., 2009; Hoose et al., 2010; Rogers et al., 2001). Most field projects sampled air parcels
in anvils of convective clouds, cirrus and winter mixed-phase stratiform clouds, keeping airborne equipment in good working
condition. However, few projects sampled air parcels through cores in convection. Thus, current observation is insufficient for
describing the whole convective cloud, especially the deep convection in severe storms. Absence about microphysical
observations of ice-nucleating particles within severe storms leads to uncertainty in understanding cold cloud process, e.g.,
hailstone formation(Li et al., 2020).
Recently, detection for soluble ions along with isotopic analysis of a huge hailstone revealed an up-and-down hailstone
growth trajectory, which demonstrated that the different shells were formed at different heights (Li et al., 2020). Further
analysis revealed large diversity in number concentration of soluble ions among hailstones from different hailstorms (Li et al.,



2018). These studies have proved aerosol information in convective cloud may be recorded in soluble particles within
hailstones(Li et al., 2020, 2018; Knight, 1981; Jouzel et al., 1975). Similarly, insoluble particles in hailstones can also record
aerosol information in severe storms.

Former studies showed that species and number concentration of insoluble particles in hailstone(Vali, 1968; Rosinski,

1966; Michaud et al., 2014) would influence heterogeneous nucleation process(Hoose and Möhler, 2012) and further hailstone
formation(Knight, 1981). Information on the species of insoluble particles can determine the freezing temperature when these
particles participate in the initiation of ice crystal formation and subsequently impact hailstone embryo growth. Biological
particles in hailstones, such as pollen and bacteria, are more efficient ice-nucleating particles than dust within the ice nucleation
region of storm clouds (Michaud et al., 2014). They can raise the freezing threshold temperature above −15 °C , while dust
particles are activated to form ice crystals at temperatures below −15 °C(Michaud et al., 2014). In addition to species, number
concentration of insoluble particles can also influence the hailstone formation. When more dust particles were considered , a
model simulation resulted in larger number concentration of ice crystals, smaller graupels (one type of hailstone embryos) size,
and suppression in the hailstone growth (Chen et al., 2019). Nonetheless, previous studies involving analysis of insoluble
particles in hailstones mainly focused on substances analysis or total number concentration statistics. A size distribution of
insoluble particles in hailstones with species information, which is beneficial for completing microphysical observation in
severe storms, has not been given so far.

This study identified insoluble particles present in hailstones, which were collected from 8 hailstorms occurred in China

between 2016 and 2021, by scanning electron microscopy (SEM) and energy dispersive X-ray spectrometry (EDX). These
insoluble particles were grouped into three species by self-organized maps (SOM) and random forest method. Variation of size
distribution of insoluble particles in embryos and different shells was explored. Based on these analysis data, logarithmic
normal distributions were fitted to describe different species of insoluble particle in deep convection.
**2 Methods**
**2.1 Sample information and experimental design**

Hailstones were collected from eight hailstorms occurring in six provinces of China during warm seasons from 2016 to

2021, and stored in clean containers, such as plastic bags, glass containers and tinfoil, by volunteers during or just after hail
(Table. 1, Fig. 1). All hailstone samples were transported to a laboratory at Peking University in Beijing and stored at
temperatures between −18°C and −4°C. The hailstones were transferred into vacuum-sealed plastic pockets and kept in a
freezer, with the internal temperature maintained between −29°C and −23°C, until further processing and analysis.


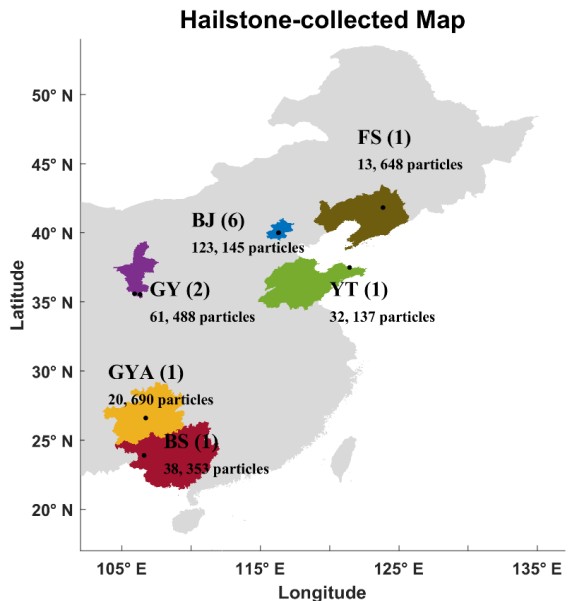

**Fig. 1: Geographical distribution of hailstone-collected provinces. Black dots are collecting locations of hailstones. Provinces of China from which the hailstones were collected are shown in different colors. The sample abbreviations are marked in the figure with number of hailstones sampled in parentheses. Abbreviations (corresponding to Table. 1): BJ, Beijing City; GY, Guyuan City; BS, Baise City; FS, Fushun City; YT, Yantai City; GYA, Guiyang City.**





| Date & Local Solar Time[a] | Location[b] | Total column water vapor[c] (kg / m²) | Freezing level height – orography[d] (m) | City & Sample abbreviation[e] | Samples[f] | Diameter[g] (mm) | Particle number[h] |
|---|---|---|---|---|---|---|---|
| 19 June 2018, 18:30 | 41.82° N, 123.85° E | 26.359[18] | 3241.66[18] | Fushun City (FS) | 1 | 13.80 | 13, 648 |
| 10 June 2016, 14:30 | 40.00° N, 116.32° E | 36.86[14] | 3780.52[14] | Beijing City (BJ1) | 1 | —— | |
| 30 June 2021, 19:00 | 39.99° N, 116.30° E | 31.73[18] | 3854.52[18] | Beijing City (BJ2) | 5 | 25.38 | 123, 145 |
| | | | | Beijing City (BJ3) | | 24.11 | |
| | | | | Beijing City (BJ4) | | 16.30 | |
| | | | | Beijing City (BJ5) | | 14.86 | |
| | | | | Beijing City (BJ6) | | 22.80 | |
| 01 Oct 2021, 14:02 | 37.49° N, 121.44° E | 32.81[13] | 3642.42[13] | Yantai City (YT) | 1 | 45.00 | 32, 137 |
| 25 Aug 2020, 17:00 | 35.53° N, 106.32° E | 17.83[16] | 422.58[16] | Guyuan City (GY1) | 1 | 15.00 | 61, 488 |
| 26 Aug 2022, 15:00 | 35.58° N, 105.93° E | 17.01[14] | 835.04[14] | Guyuan City (GY2) | 1 | 18.50 | |
| 14 Apr 2016, 19:00 | 26.60° N, 106.72° E | 31.62[18] | 2147.58[18] | Guiyang City (GYA) | 1 | 26.20 | 20, 690 |
| 09 May 2016, 17:51 | 23.90° N, 106.60° E | 47.45[17] | 4572.70[17] | Baise City (BS) | 1 | —— | 38, 353 |

**Table. 1: Information about collected hailstones.**

[a]  **Date and local solar time of hailstorms occurrence. Hailstones were collected within 30 min during hail.**
[b]  **Hailstone collecting location.**
[c]  **The total column water vapor values (local solar time of ERA5 reanalysis data in square brackets**(Hersbach et al., 2018)**).**
[d]  **Depth between freezing level height and orography (local solar time of ERA5 reanalysis data in square brackets**(Hersbach et al., 2018)**).**
[e]  **Sample abbreviations.**
[f]  **Numbers of hailstones used in experiments.**
[g]  **Diameter of hailstone (——  means no record).**
[h]  **Insoluble particle number in hailstones.**



Insoluble particles were extracted in the experiments (Fig. 2). The surface of the hailstone was polished to remove any
attached grass or soil. Then, hailstones were sliced into cross-sections along the major axis, corresponding to the size of the
hailstone embryo. The cross-sections were sliced into several shells using heated Fe-Cr alloy wire at air temperature below
−8°C. The shells within a hailstone were distinguished based on their natural transparency or opacity. Hailstones with a major
axis < 7 mm could not be sliced because of the mass loss with heating using our experimental apparatus.

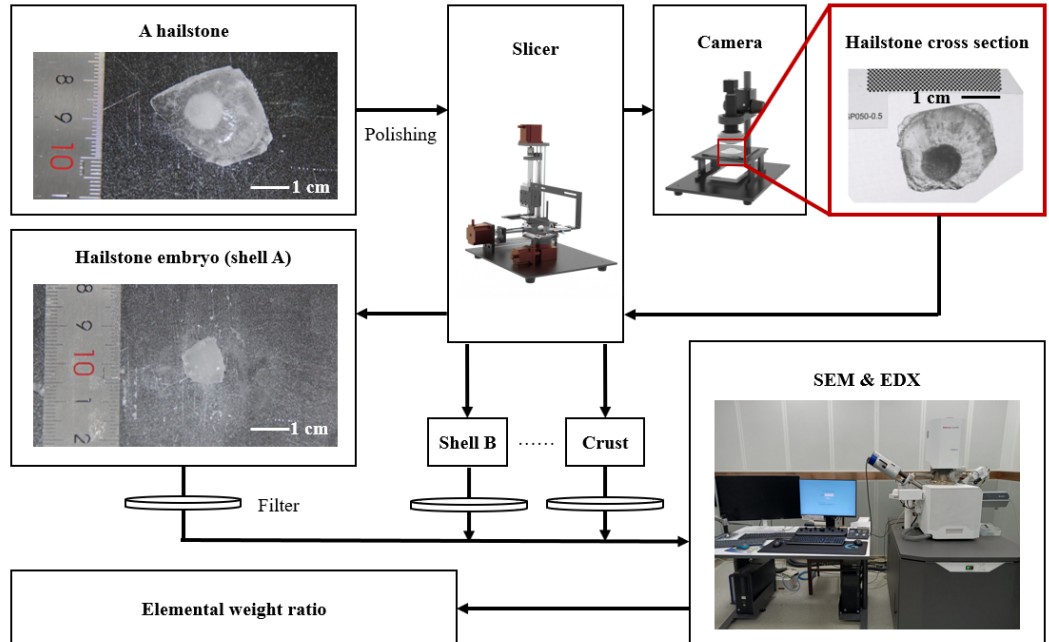


**Fig. 2: Schematic diagram showing the experimental framework. The surface of the hailstone was polished to remove any attached**
**grass or soil and sliced into cross-sections along the major axis. The shells within a hailstone were distinguished based on their**
**natural transparency or opacity. Solution of melting shell samples run through a filter membrane to obtain insoluble particles. Each**
**shell sample was analyzed within about 4 hours by scanning electron microscopy and energy dispersive X-ray spectrometry for**
**elemental weight ratios of insoluble particles.**

The shells were labeled with capital letters in alphabetical order from the inner shell to the crust. For example, the embryo
of a hailstone was designated as shell A. To obtain insoluble particles, the shells were melted into solution, and run through a
filter membrane (VSWP01300, Merck KGaA, Germany) with a pore size of 30 nm. The filter membrane was flushed five
times with 1 mL of distilled water to ensure as many insoluble particles as possible stuck on the filter membrane. The filter
membrane was dried under an air temperature of about 40°C for electron microscopy requirements.
The number of insoluble particles in each shell was determined by scanning electron microscopy (SEM), focusing on
particles > 0.16 μm. The length along the major axis of particles was measured using Aztec software (Aztec software, Oxford



Instruments plc, UK) on SEM images. Energy dispersive X-ray spectrometry (EDX) was used to determine the elemental
weight ratios of particles. Only elements with an atomic number > 4 could be detected because the X-ray input window was
made of beryllium. Each shell sample was analyzed within about 4 hours by SEM and EDX. The scanning mode of SEM was
set in random order to reduce the error caused by bias in detection area.
**2.2 Clustering and classification**
The number of insoluble particles was measured using Aztec on SEM images, but the species could not be determined
directly and were identified by machine learning. The criteria of species classification were established by the self-organized
maps method to determine the species of unclassified particles. These labeled particles were then regarded as true species and
used to train a random forest classifier. Details are presented in Fig. 3.

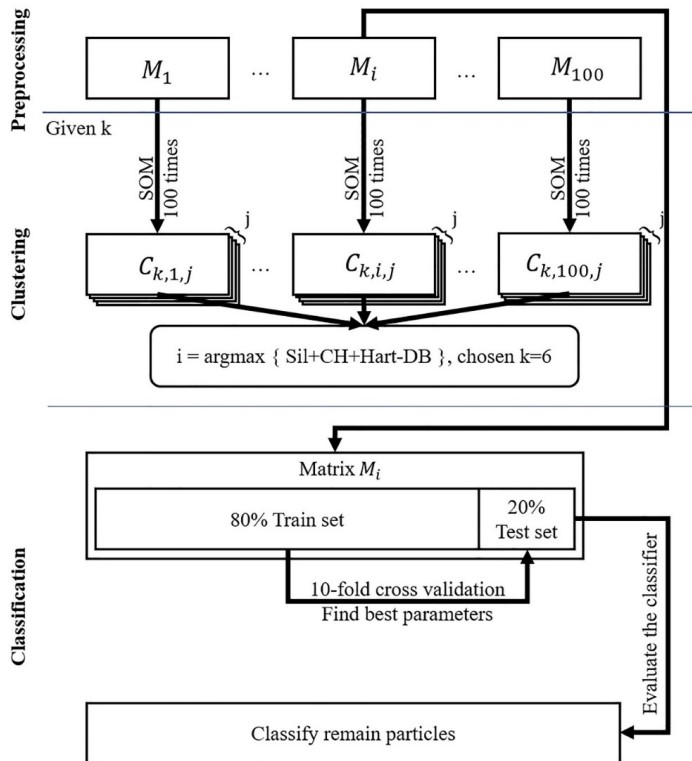


**Fig. 3: Schematic diagram showing the framework of the methods used for particle identification in this study. The 100 matrices $M_i$**
**with $i$ ranging from 1 to 100 were used in self-organized maps clustering analyses, and each of them included unidentified 81,888**
**particles with 19 elemental features (N, Na, Mg, Al, Si, P, S, Cl, K, Ca, Ti, Cr, Mn, Fe, Ni, Cu, Br, Ba, and Pb). Centroid matrix $C_{k,i,j}$**
**was clustering results by self-organized maps method with chosen cluster number $k$. The operation of self-organized maps with the**
**same $k$ was repeated 100 times to ensure the robustness of results. The $j$ is the number of repeating time, ranging from 1 to 100.**
**Four indexes, i.e., Silhouette index(Sil), Calinski–Harabasz index (CH), Hartigan index (Hart), and Davies–Bouldin index (DB) were**



used to determine best centroid number $k$, $i$, and $j$. $\mathbf{M_i}$ containing identified 81,888 particles, was separated as training and test
set in random forest classification with 10-fold cross validation. The best classifier was used to classify remain particles.

With reference to the studies of Ault et al. in 2012 and Kirpes et al. in 2018 and considering the results of elemental
weight ratios determined by EDX analysis (Ault et al., 2012; Kirpes et al., 2018), 19 elements (N, Na, Mg, Al, Si, P, S, Cl, K,
Ca, Ti, Cr, Mn, Fe, Ni, Cu, Br, Ba, and Pb) were selected to confirm the species of particles. C and O were not taken in account
when clustering or classifying particles as the membrane filters were made from cellulose acetate and cellulose nitrate, which
contain C, H, N, and O. We could not detect H because the ray-input window was made of beryllium. All particles showed
high contents of C and O but different contents of N, so N was retained as a feature of classification.
Species of aerosol particles vary regionally(Tao et al., 2017). Therefore, when establishing the matrices of elemental
weight ratios for clustering, equal amounts of data were randomly extracted from the sample data from each province to ensure
the inclusion of a consistent proportion of samples from each region in the training process. A hailstone FS from Fushun City,
Liaoning Province was shown to contain 13,648 insoluble particles, which was the smallest among all samples from six
province (Fig. 1). With random sampling of 13,648 particles from each province, the matrix used in clustering analyses
included 81,888 particles. This operation was repeated 100 times to obtain 100 matrices $\mathbf{M_i}$ with $i$ ranging from 1 to 100.
Each matrix $\mathbf{M_i}$ was clustered using the SOM method, which is an unsupervised machine learning method that
represents high-dimensional data in low-dimensional space while preserving the topological structure of the data. The neuronal
network was set to $k$ neurons in a layer, where $k$ is the given clustering center number from 2 to 10. Each SOM operation
produces a centroid matrix $\mathbf{C_{k,\,i,\,j}}$, where $i$ is the number of particle sample replicates, as mentioned above, and $j$ is the
number of rounds of SOM operation. Weights of a neuron describe its position in multivariate space and can be taken as a
cluster center. The operation of SOM with the same neuronal network setting was repeated 100 times to ensure the robustness
of the centroid matrix $\mathbf{C_{k,\,i,\,j}}$. Four indexes, i.e., Silhouette index(Rousseeuw, 1987), Calinski–Harabasz index(Calinski and
Harabasz, 1974), Hartigan index(Sibson and Hartigan, 1976), and Davies–Bouldin index(Davies and Bouldin, 1979), were
selected as evaluation indicators to determine the parameters $k$, $i$ and $j$. The best $k$, $i$ and $j$ was chosen by combining the
evaluation of the four indexes (Fig. 4) and elemental weight ratios of each centroid.

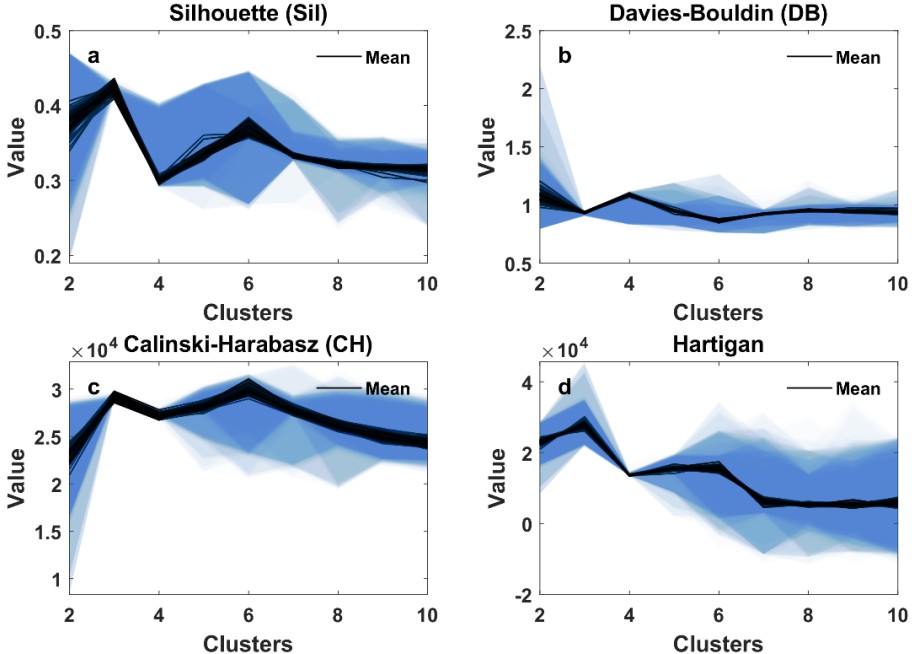

**Fig. 4: Evaluation of self-organized maps clustering results. Evaluation of self-organized maps clustering results by (a) Silhouette index, (b) Davies–Bouldin index, (c) Calinski–Harabasz index, and (d) Hartigan index. Self-organized maps operation was repeated 100 times to obtain each randomly sampled matrix $\mathbf{M_i}$. The solid lines and shading represent the average and spread of 100 repetitions, respectively.**

The centroid matrix $\mathbf{C_{k, i, j}}$ with best $k$, $i$ and $j$ was treated as a training set for random forest classification. The chosen centroid matrix $\mathbf{C_{k, i, j}}$ with the top four elements is shown in Fig. 5 with $k = 6$. The first species with low elemental weight ratio except C and O contents was considered to be organics. The second species with high Fe content and low Cr content was introduced by the material of the slicer used in the experiment. The third species had a high Al content representing oxides or carbonates of aluminum. The fourth and fifth species were mineral silicates. So that, the third, fourth, and fifth species were referred to as "dust". The last species with high N content was protein-containing biological aerosol.

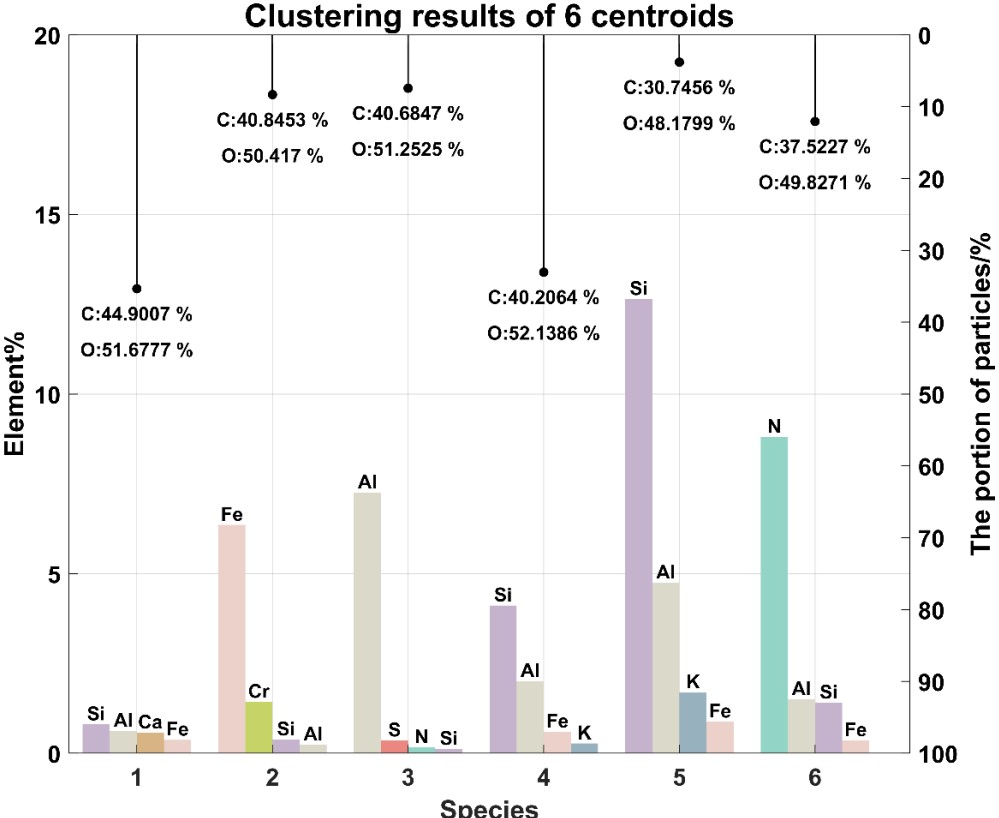

**Fig. 5: Centroids of clustering with six clusters from self-organized maps results and each species portion. Colored bars show the top four elements of each species. The stem bars show the portion of each species. The average contents of C and O of each species are marked at the end of the stem bars.**

The random forest method was applied in classifying insoluble particles, which involves randomly growing 100 classification trees. The training set consisted of 80% of $M_i$ and 10-fold stratified cross-validation was applied during the training process to find the best among the 100 random trees. The remaining 20% of $M_i$ was used as the test set to evaluate the classifier. The best classification tree and the confusion matrix of the evaluation of testing are shown in Fig. 6. All remaining insoluble particles were classified by this tree. Finally, results identified three species: organics, dust, and bioprotein aerosols.






**Fig. 6: Confusion matrix of the best random forest classifier tree. The numbers on the diagonal are accurately predicted insoluble**

**particles. Numbers in bold indicate the accuracy of prediction of each type.**


**2.3 Calculation of insoluble particles number concentration**
Particle number was converted to a number concentration as follows:

$$n_{liquid} \cdot V_{liquid} \; = \; N_{liquid} \; = \; N_{dilute} \; = \; n_{diluted} \cdot V_{diluted} \tag{1}$$

Part of the solution was not consumed in these experiments and was retained as a backup. During several experiments,
the melted shell solution was diluted.

$$n_{diluted} \; = \; n_{used} \; = \; \frac{N_{used}}{V_{used}} \; = \; \frac{N_{filter}}{V_{used}} \tag{2}$$

SEM can provide the number of particles on a filter, but the whole area of the filter cannot be scanned. We assumed that
the particles were uniformly distributed on the filter and the scanning mode of SEM was set as "random scanning". A such
relasionship between the number of scanned particles and the number of particles on the filter:

$$\frac{S_{filter}}{S_{images}} \; = \; \frac{N_{filter}}{N_{count}} \tag{3}$$



In the above formulas, $n$ is the number concentration of insoluble particles; $N$ is the number of insoluble particles; $V$
is the volume of the solution; $S$ is the area of the filter; subscript *liquid* refers to the melted shell; subscript *diluted* refers to
the diluted solution; subscript *used* refers to the consumed diluting solution; subscript *filter* refers to the filter membrane;
$N_{count}$ is the number of particles counted on the filter; and $S_{images}$ is the area of the microscopic image.
These three formulas were reduced to Eq. (4):
$$n_{liquid} = \frac{1}{V_{liquid}} \cdot \frac{S_{filter}}{S_{images}} \cdot \frac{V_{diluted}}{V_{used}} \cdot N_{count} \tag{4}$$

where $S_{filter}, S_{images}, N_{count}, V_{diluted}$, and $V_{used}$ can be measured. The liquid volume was the mean of readings by
two experimenters from the test tube calibration. From Eq. (4), a tiny change in $n_{liquid}$ can be expressed as $dn_{liquid}$:
$$dn_{liquid} = n_{liquid} \cdot \left( -\frac{dV_{liquid}}{V_{liquid}} + \frac{dV_{diluted}}{V_{diluted}} - \frac{dV_{used}}{V_{used}} + \frac{dN_{count}}{N_{count}} \right) \tag{5}$$

As,
$$dS_{filter} = dS_{images} = 0 \tag{6}$$

The uncertainty comes from the measurement error of the experimental instruments.
$$\Delta = |dn_{liquid}| \leq n_{liquid} \cdot \sqrt{\left(\frac{dV_{liquid}}{V_{liquid}}\right)^2 + \left(\frac{dV_{diluted}}{V_{diluted}}\right)^2 + \left(\frac{dV_{used}}{V_{used}}\right)^2 + \left(\frac{dN_{count}}{N_{count}}\right)^2} \tag{7}$$

So,
$$\Delta_{max} = n_{liquid} \cdot \sqrt{\left(\frac{dV_{liquid}}{V_{liquid}}\right)^2 + \left(\frac{dV_{diluted}}{V_{diluted}}\right)^2 + \left(\frac{dV_{used}}{V_{used}}\right)^2 + \left(\frac{dPs}{Ps}\right)^2} \tag{8}$$

Here, the minimum scale of the test tube containing melting solution is 0.1 mL and $dV$ is the greatest reading error
caused by human and was set to 0.05 mL. $\frac{dN_{count}}{N_{count}}$ represents the uncertainty of detecting insoluble particles, which is related
to the scan settings.
$$\frac{dN_{count}}{N_{count}} = \frac{dPs}{Ps} = \frac{3}{6,340,608} \tag{9}$$

$dPs$ is the minimum number of pixels that can be detected in an image. $Ps$ is the total number of pixels in the
micrograph.
**2.4 Curves fitting**
We aggregated our data into 0.2-µm intervals (e.g., particle number concentration at $D = 0.3$ µm, corresponding to the
sum of particles of diameter 0.2–0.4 µm) to fit the logarithmic normal distribution:
$$n(\ln D) = \frac{N}{\sqrt{2\pi} \ln \sigma_g} \cdot \exp\left[ -\frac{(\ln D - \ln r_g)^2}{2 \ln^2 \sigma_g} \right] \tag{10}$$






$$n(D) = \frac{1}{D} \cdot n(\ln D) \tag{11}$$

Here, $n(\ln D)$ and $n(D)$ are the size distributions of particles, $D$ is the diameter of insoluble particles, and $N$ is the

total number concentration of particles. According to the above, when the $N_{count}$ in an interval equals 1, the number
concentration will show a flat tail because of the conversion to obtain $n_{liquid}$. The fitting data were selected with intervals
equals to 0.2 μm. The least squares method was applied to determine the fitting parameters and $R^2$ was used to estimate fitting
parameters. The two centroids of fitting parameters of organics and dust were determined by K-means method.
**3 Results**

Total 289,461 insoluble particles from 30 shells of 12 hailstones were detected by scanning electron microscopy.

Elemental weight ratios of each particle were determined using energy dispersive X-ray spectrometry. More details regarding
calculating number concentration of insoluble particles per cubic centimeter volume water (hereinafter referred to as number
concentration) from number of insoluble particles were showed in method description. Identification of insoluble particles
used self-organized maps for clustering and random forest for classification. Four indexes were selected to determine the
appropriate parameters of clustering. The clustering results were set as training and testing set of classification. A confusion
matrix of the classifier showed that the accuracy, precision, and recall were 99.7%, 99.4%, and 99.5%, respectively. All
particles were identified as organics, dust, and bioprotein aerosols (i.e., the fraction of biological aerosols with protein content).
**3.1 Sample representativeness**

Five of the 12 hailstones (BJ2–BJ6) were from the same hailstorm that occurred in Beijing on June 30, 2021. The insoluble

particles present in these hailstones showed similarity in the size distribution of organics, dust, and bioprotein aerosols but
differed from other 7 hailstones that from other hailstorms (Fig. 7). The results were similar to those of Li et al., who reported
that the number concentrations of water-soluble ions varied among hailstorm events but showed similarity in the same storm
(Li et al., 2018). These analyses suggested that insoluble particles in the hailstorm may come from local natural or
anthropogenic emissions (e.g., soil dust, aerosols from biomass and fossil fuel combustion, products of the conversion of
gaseous precursors), which is also suggested by the results on water-soluble ions (Beal et al., 2022). The updraft within the
hailstorm is likely to bring insoluble particles from local surfaces or boundary layers into deep convective clouds, as hailstorms
are among the most severe storms with strong updrafts (Battaglia et al., 2022). BJ2 was selected to represent five hailstones
from the same hailstorm in further analysis to simplify comparison.



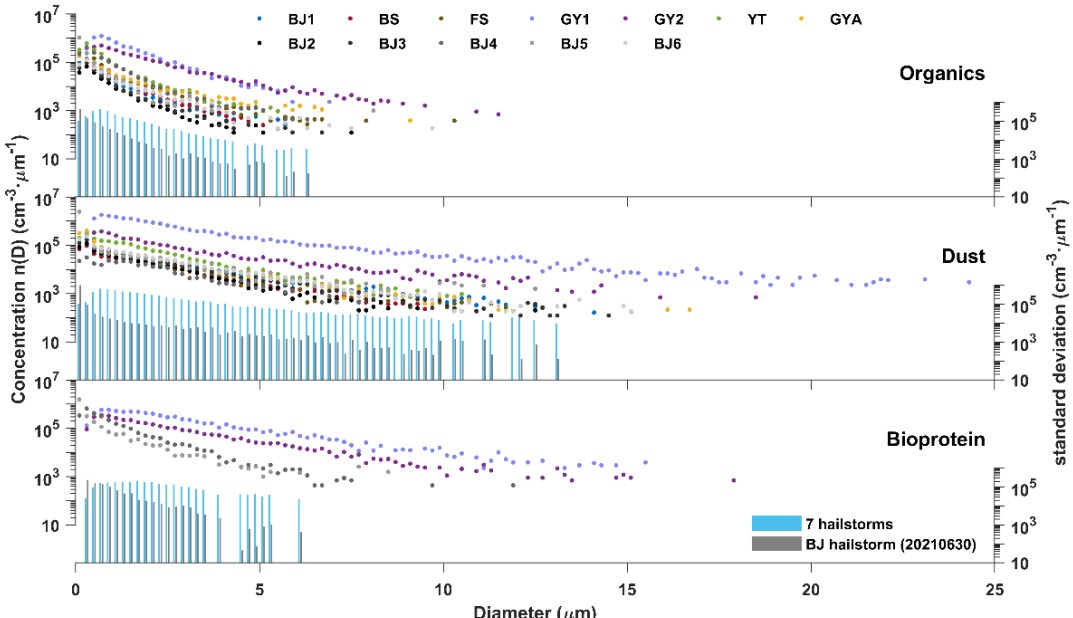

**Fig. 7: Size distribution of organics, dust, and bioprotein aerosols of insoluble particles in 12 hailstones. Each number concentration at diameter D total number concentration of insoluble particles with diameter ranging from D − 0.1 µm to D + 0.1 µm. Colored dots refer to seven hailstones (FS, BJ-1, YT, NX-1, NX-2, GY, and BS) from seven different hailstorms. Black and gray dots refer to five hailstones (BJ-2–BJ-6) from the same hailstorm that occurred in Beijing on June 30, 2021. Blue and gray bars show the standard deviation of insoluble particles from seven hailstorms and one hailstorm, respectively. Abbreviations (corresponding to Table. 1): BJ, Beijing City; GY, Guyuan City; BS, Baise City; FS, Fushun City; YT, Yantai City; GYA, Guiyang City.**

### 3.2 Size distribution in embryos

All hailstone embryos analyzed in this study were graupels, which grows from the initial ice particles through accretion of supercooled droplet (Knight, 1981). These initial ice particles are likely formed by insoluble particles where heterogeneous nucleation processes (Lamb and Verlinde, 2011). That is, insoluble particles in graupels possibly affected the formation of ice crystals and subsequently affected the formation of hailstone embryos. The size distributions of insoluble particles in eight hailstone embryos (BJ1, BJ2, GY1, GY2, BS, FS, YT, and GYA) were shown in Fig. 8.



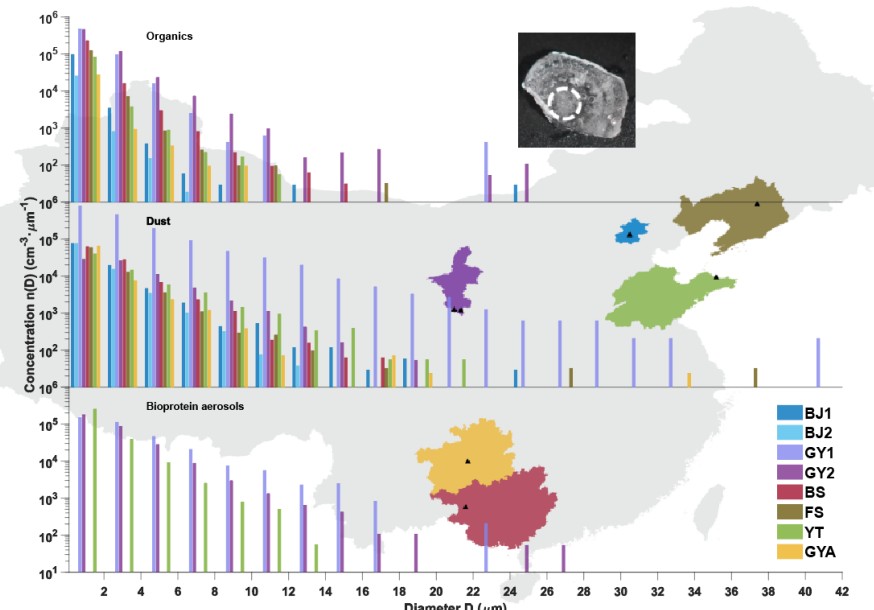

**Fig. 8: Size distribution of insoluble particles in hailstone embryos. Provinces of China, from which the hailstones were collected, are shown in different colors. Black triangles indicate the locations of hailstone sample collection. The white dashed circle shows part of the hailstone embryo. Abbreviations (corresponding to Table. 1): BJ, Beijing City; GY, Guyuan City; BS, Baise City; FS, Fushun City; YT, Yantai City; GYA, Guiyang City.**

As mentioned above, BJ2 represented BJ2–BJ6. The variations in number concentrations of dust and bioprotein insoluble particles indicated that particle number concentrations decreased exponentially with particle diameter, with marked variation observed among hailstorms. The distribution distinguished organics from dust and bioprotein aerosols as the number concentrations of organics from all samples decreased with particle diameter before 8 μm, while those of GY1 and GY2 fluctuated starting at diameters of 8 μm and 12 μm, respectively. This was likely due to some uncontrolled residential and industrial coal burning in GY (Guyuan City). A great variance existed in size distribution of both organics and dust. The number concentrations of organics from a hailstone embryo were 1 to 390 times to those from different hailstone embryos at the same diameter. The number concentrations of dust from a hailstone embryo were 1 to 527 times to those from different hailstone embryos at the same diameter. The number concentrations of dust from BJ1, BJ2, and GY1 were at least 3 times higher than organics in particles of the same diameter in the range of 2–24 μm. Moreover, dust showed a wider size distribution than organics and bioproteins among all samples, since dust from GY1 had a higher number concentration and larger maximum size (42 μm) than from other hailstone embryos. Bioprotein aerosols, with high freezing efficiency, may have formed initial ice particles in GY1, GY2, and YT, while dust or organics caused initial ice particle formation in hailstorms in cases lacking bioprotein aerosols. All hailstone embryos contained organics and dust, but not all hailstone embryos contained a significant



amount of bioprotein aerosols. There were uncertainties in quantification of biological aerosols, due to poor understanding of
biological transport and transformation processes (Fröhlich-Nowoisky et al., 2016).
**3.3 Variation in hailstone shells**

Size distribution of each species differed little in characteristics in outer shells with the embryos (Fig. 9). For a four-shell

hailstone, the number concentrations of insoluble particles showed V-shaped (BS and YT) or inverse V-shaped (BJ1)
distributions from embryo to crust. Five of nine two-shell hailstones showed higher number concentrations of dust in crusts
than embryos, while seven of them showed higher number concentrations of organics in embryos than crusts. However,
quantification of the differences in number concentration varied little among shells. The 90.5% points showed that differences
in number concentration of the same kind particles in a shell compared to the previous shell at the same diameter was within
twice, and the maximum differences was up to 9 times (294 data points in Fig. 9). This was because the growth of hailstones
beyond the embryo stage depends on the accretion of supercooled water rather than ice crystals (Lamb and Verlinde, 2011).
The hailstone recorded not only insoluble particles when the embryo formed, but also insoluble particle in the hailstone growth
zone throughout the hailstorm. Thus, the size distribution of particles within the whole hailstones may represent the distribution
of insoluble particles in deep convection where the hailstones pass through.

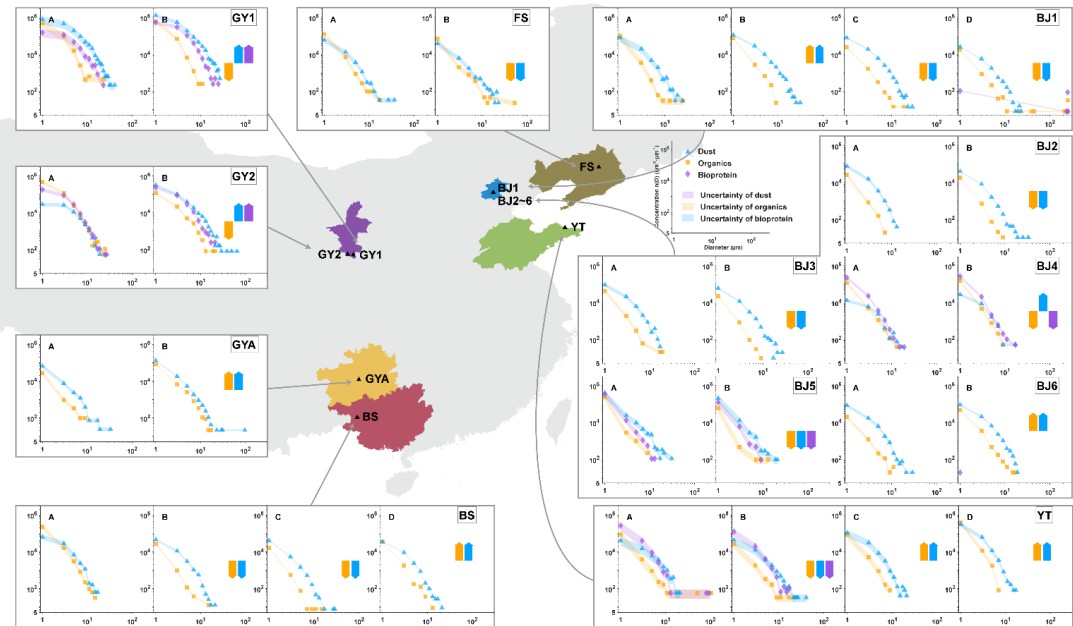


**Fig. 9: Size distribution of insoluble particles present in natural shells of 12 hailstones. The diameter interval on the x-axis is 2 μm.**
**The y-axis shows the particle number concentration from $D - 1$ μm to $D + 1$ μm. Blue triangles, orange rectangles, and purple**
**diamonds indicate dust, organics, and bioprotein aerosols, respectively. The natural shells were named alphabetically with capital**



letters (shell A refers to embryos and shell B/D refers to crust of hailstones). The arrow direction indicates the trend of particle
number concentration in this layer with regard to the previous layer. Uncertainty is indicated by shading. Calculations are described
in detail in supplementary information. Abbreviations (corresponding to Table. 1): BJ, Beijing City; GY, Guyuan City; BS, Baise
City; FS, Fushun City; YT, Yantai City; GYA, Guiyang City.

**3.4 Logarithmic normal distribution of dust and organics**

The size distributions of dust and organics in the whole hailstone can be described by a logarithmic normal distribution

(Fig. 10a) (Lamb and Verlinde, 2011):
$$n(\ln D) = \frac{N}{\sqrt{2\pi}\ln\sigma_g} \cdot exp\left[-\frac{(\ln D - \ln r_g)^2}{2\ln^2\sigma_g}\right], (D > 0.2\ \mu m) \tag{12}$$

Where $n(\ln D)$ is the number concentration of insoluble particles per cubic centimeter volume water ranging from

$\ln D - \frac{1}{2}\mathrm{d}\ln D$ to $\ln D + \frac{1}{2}\mathrm{d}\ln D$. Here, $D$ represents the diameter of particles (in micrometers), $\ln r_g$ is the geometric
mean diameter, and $\ln\sigma_g$ is the geometric standard deviation (Lamb and Verlinde, 2011). The number of bioprotein aerosols
was below the limit of detection in some samples, so that, only the curves of organics and dust were fitted. The fitting
parameters of the same species were aggregated in parameter space, and were suspected to be related to the physical properties
of each species, requiring further studies for confirmation. Moreover, the fitting parameters of organics and dust particles were
clustered into two centroids (Fig. 10b) by the K-means method, which indicated that organics and dust have two classic modes
(classic mode of organics: $\ln r_o$ = -0.70 μm, $\ln\sigma_o$ = 0.91 μm, and $N_o$ = 9.19 × 10⁵ cm⁻³; classic mode of dust: $\ln r_d$ = 0.11
μm, $\ln\sigma_d$ = 1.07 μm, and $N_o$ = 1.58 × 10⁶ cm⁻³). That is, insoluble organics in hailstones are usually smaller in diameter and
present in lower amounts than dust. Regardless of fine or coarse particles ($D$ < 0.5 μm in diameter were not considered in
reference to DeMott et al. (DeMott et al., 2010)), the number concentration of dust was up to 2 orders of magnitude higher
than the number concentration of organics. These observations indicated that dust accounted for the major portion of particles
in eight hailstorms (no considering about bioprotein), which was consistent with the observations of embryos described above.



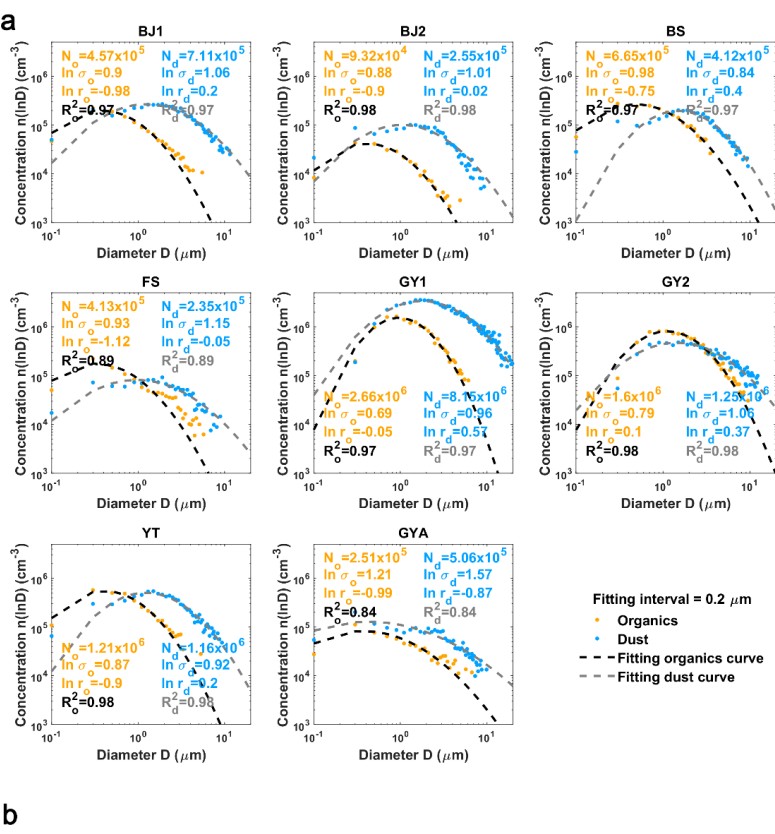

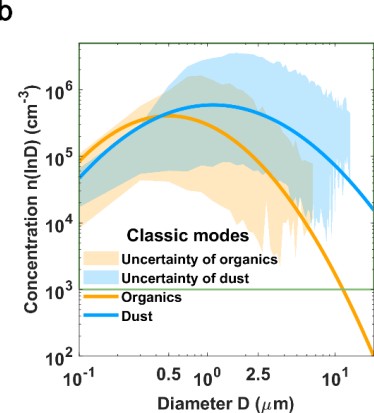


Fig. 10: Fitting size distribution functions of organics and dust contained in the whole hailstone. (a) Fitting parameters of logarithmic

normal distributions of BJ1, BJ2, BS, FS, GY1, GY2, YT, GYA. (b) Classic modes of dust and organics (interval of data is 0.2 μm

and fitting curves painted with interval of 0.02 μm). The fitting range of (a) is shown with a green rectangle. The centroid of the

organics fitting parameter (orange line) is $\ln \sigma_o = 0.91$ μm, $\ln r_o = -0.70$ μm, and $N_o = 9.19 \times 10^5$ cm$^{-3}$. The centroid of the dust

fitting parameter (blue line) is $\ln \sigma_d = 1.07$ μm, $\ln r_d = 0.11$ μm, and $N_d = 1.59 \times 10^6$ cm$^{-3}$. Shading showed uncertainty of organics

and dust. Abbreviations (corresponding to Table. 1): BJ, Beijing City; GY, Guyuan City; BS, Baise City; FS, Fushun City; YT, Yantai

City; GYA, Guiyang City.



**4 Conclusions**

This was the first study to simultaneously analyze both the number concentrations and species (organics, dust and bioproteins) of insoluble particles in hailstones. Analysis of insoluble particles present in hailstones, which participate in heterogeneous nucleating process as ice-nucleating particles in a deep convection(Lamb and Verlinde, 2011), provides a new approach for refinement of particle observation in severe storms and the understanding of hailstone formation.

The size distribution of insoluble particles in hailstones from the same hailstorm showed less variation than those from different hailstorms. One possible reason is that updrafts of hailstorms brought insoluble particles from local surfaces or boundary layers into deep convective clouds. Moreover, part of these insoluble particles participate in freezing initial ice particles to form one type of hailstone embryos. Almost all insoluble particles in hailstone embryos analyzed in this study showed an exponential size distribution, which was consistent with the effects of gravity. The number concentrations of organics and dust from different hailstone embryos differed up to 389 times and 526 times at the same diameter, respectively. Hailstone samples with high insoluble particle content, i.e., GY1 and GY2, showed significantly lower total column water vapor values and smaller depth between freezing level height and orography within one hour before hailstorm occurrence, compared to other samples (Hersbach et al., 2018). The competition of condensation and shorter updraft pathway might be responsible for the high number concentrations of organics, dust, and bioproteins in GY1 and GY2. Size distribution of insoluble particles varied in shells up to 9 times, which was much small than differences with different hailstorms.

Two classic size distribution modes of organics and dust in hailstones were fitted as logarithmic normal distribution for description of insoluble particles in deep convection where the hailstones grew up. The two classic size distribution modes of insoluble particles suggested that dust occupied the major fraction without taking bioprotein into account. Besides, there is a positive correlation between the number concentrations of insoluble particles and ice-nucleating particles in hailstones for corresponding species (Ren et al., 2023, submitted, figure not shown). Further measurement of ice-nucleating particles by drop-freezing experiments will establish the relationship between insoluble particles and ice-nucleating particles. Combination of these results with future experiments to determine the number concentrations and species of particles from local observations will establish the relationship between surface observation and ice-nucleating particles in deep convective clouds, which will lead to improvement of the parameterization of ice-nucleating particles in both weather and climate models.

However, two kinds of classic size distribution modes of organics and dust in hailstones were performed, but a more robust classic mode required a larger number of samples. In future, for any climate or weather model, the classic mode can be assumed as the mean state to describe the characteristics of insoluble particles in supercooling water. In addition, this study did not attempt to parameterize bioprotein aerosols, because there was a great uncertainty in quantification due to poor understanding of biological processes(Fröhlich-Nowoisky et al., 2016). Further collaborative studies are required to gain a better understanding of biological processes to establish the classic bioprotein mode.



**Code availability**

Self-organized maps algorithm is functions on MATLAB

https://ww2.mathworks.cn/help/deeplearning/ref/selforgmap.html

Random forest algorithm is functions on MATLAB

https://ww2.mathworks.cn/help/stats/treebagger.html?searchHighlight=TreeBagger&s_tid=srchtitle_TreeBagger_1

The 10-fold stratified cross-validation algorithm is functions on MATLAB

https://ww2.mathworks.cn/help/stats/cvpartition.html?searchHighlight=cvpartition&s_tid=srchtitle_cvpartition_1

Identification algorithms are coded on MATLAB and will be made available on request.

**Data availability**

Data will be made available on request.

**Author contributions**

Haifan Zhang wrote the original draft under the concept presented by Qinghong Zhang. Haifan Zhang, Xiangyu Lin and Chan-Pang Ng participated in preprocess and reservation of hailstones from volunteers. Haifan Zhang and Xiangyu Lin sliced hailstones using machine manufactured by Kai Bi and performed the experiments on analyzing element weight ratio of insoluble particles with help of Li Chen. Kai Bi also provided hailstones BJ2 ~ BJ6. Machine learning on identification of particles is operated by Haifan Zhang. Yangze Ren and Huiwen Xue compared ice nucleation particles from drop-freezing experiments with our data. Zhuolin Chang provided hailstones GY1 and GY2. All authors discussed and contributed to the final manuscript. Qinghong Zhang directed this project.

**Competing interests**

The authors declare no competing interests.

**Acknowledgments**

This study was supported by the National Natural Science Foundation of China (Grant Nos. 42030607 and 41930968), the Innovation Project of the China Meteorological Administration (Grant No. CXFZ2021J038) and the Key R & D projects in Ningxia Hui Autonomous Region (2022BEG02010). The authors thank Cai Yao from the Meteorological Bureau of Guangxi, China in collecting hailstones BS in Guangxi. The authors thank volunteers in collecting hailstones. The authors thank Prof. Jiwen Fan from Pacific Northwest National Laboratory of the United States for discussions.



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
