# Peer review of "Analysis of insoluble particles in hailstones in China"

_EGUsphere, 2023_

## Referee Comment (RC2)

egusphere-2023-290
Recommendation: Accept pending major revisions

This article provides some of the first observations of insoluble particle concentrations within hailstones collected at the ground, as well as the opportunity for embryo analysis. My primary concern is that while this observational dataset is unique and worthwhile, the authors do not use it to draw any conclusions about physical processes related to hailgrowth. It isn't clear what the purpose of these observations is. The description of the classification and clustering technique is also confusing. Once these issues are addressed I am comfortable with publication.

Major Comments:

1. The authors have clearly spent a lot of effort to gather these observations, and they are some of the first studies of insoluble particles in hailstones. However, I would like to see some conclusion about the physical processes that these observations can now help us to understand. What, physically, can we now conclude about hailstone growth, or insoluble particles in hailstones, that we couldn't before? As it stands, the article is simply a reporting of what the authors found. What do these observations *mean*?

What information does the knowledge about how the particle distributions change among embryos/shells provide about physical processes involved in hailstone growth? For example:
   o   Which hailstones had size distributions that changed significantly between one shell to the next? Why might this occur?
   o   Do shell B and embryo distributions usually look the same or different? Shell C and B? Do size distribution/particle type vary more among shells, or among hailstones? Why?
   o   Do you have a sense if these hailstones might have taken similar paths through the storms that generated them?
   o   Were the storms that generated these hailstones all of similar convective mode, or different? Would you expect storms with stronger updrafts to transport more particles, changing the insoluble particle size distribution? Why or why not?

2. The description of the classification and clustering techniques in Section 2.2 is very difficult to understand, making the subsequent results in the rest of the paper unnecessarily hard to follow.
   •   Fig 2: This figure is hard to follow. I'd label each step with a single action (e.g., polishing, slicing, shell extraction, etc.), and then list and describe each action by name in the caption (or in the text itself) Numbering the steps and boxes would also help. That way it is clear which stage of the process corresponds to which box in the figure. Names of equipment can go below each image, in smaller font perhaps.
   •   Lines 126- 127: This step should be included as a step in Fig. 3, showing what M1 - M100 are and how they are generated.
   •   Lines 128- 137: The SOM description could be clearer. What is being produced by this process? Identification of particle species? How is that determined from a given centroid matrix? Are the number of outcome clusters predefined (a perhaps that is k?) If so, how are the numbers k-2 through 10 selected?
   •   Lines 133-134: Does the "same neuronal network setting" mean k is the same, or k, i, and j are all the same? For that matter, what does "neuronal network setting" mean?
   •   What are "particle sample replicates"?
   •   Lines 134-136: What information does each of these indices provide? Most importantly, how do you determine accuracy with this method?

- Lines 144- 149, Fig. 5: Unfortunately, I can't follow this description at all. What information does a centroid matrix have in it? Are carbon and oxygen included in the classification matrix, and as SOM inputs, or not? Lines 118- 121 seem to indicate no.
- Lines 144 -159: Please explicitly state what is being used as the "truth" dataset for the random forest. I had thought it would be the centroid matrix (line 144), but instead I think it is the classification outcome of the centroid matrix (Fig. 6)?

3. The number of studies of observed hailstone embryos in the literature less than 30 years old is almost zero, so it seems a real missed opportunity not to offer some details about the embryos collected here. These were all graupel embryos, not frozen drops? How do the characteristics of these embryos and hailstorms correspond to the hailstone embryo research of Knight (1981, J. Appl. Meteorology and Climatology)? How big were each of the embryos? Are you able to estimate their density? Differently sized embryos would also have to have impacts on their insoluble particle makeup, I would think. Do you find that to be the case?

Minor Comments:
- Lines 170- 173: I think I get your meaning here, but it should be clearer. Do you mean that because you assume that the random subsample of the filter is representative of the entire filter, Ncount is determined by multiplying the observed Nfilter by the area ratio between the whole filter and the obsered image (Simages/Sfilter)? If so, I would explain it like that.

- Lines 174-177: Move these sentences to the start of the subsection immediately following (1). Also, adding a sentence after each equation explaining the physical meaning of it would be helpful In the reader. E.g., The number of insolvable particles in the melted shell (Nliquid) can be found by multiplying their number concentration (nliquid) by the volume of the melted shell (Vliquid); this total particle number does not change when the solution is diluted (Ndilute).

- Line 180: How is Ncount determined? I thought only Nfilter could be observed.

- Lines 185- 188: How are these equations determined?

- Line 196: What data? Particle number concentrations (nliquid)? Binned by particle diameter size?

- Lines 200 -203: Why was a log-normal distribution chosen? What are rg and Tg? Why this form of the distribution? What does line 202-203 mean physically?

- Lines 216- 225, Fig. 7: Given the log scale and the small y-axes of Fig. 7, it is difficult to see the differences between any hailstones, let alone among specific storms. I recommend shifting the standard deviation results to a new figure. I would also be curious to see what the standard deviation values are across all storms but excluding the GY1 and GY2 hailstones. I am curious how much of the increased standard deviation for all 7 hailstorms in sum is due to those 2 storms. Once those two storms are removed, that should have an impact on the conclusions in Lines 216- 225. Possibly, using just one Beijing hailstone is not representative.

- Lines 237 - 238: That's a lot of "possiblies". One could just as easily argue the insoluble particles were contributed by the riming supercooled water acting to form the embryo.

- Lines 248-251: I'm having trouble following these sentences. Why would industrial coal burning result in an increased number of organic aerosols specifically 10 microns in diameter?

- Lines 252- 253: Rephrase to make it clear "at the same diameter" refers to particle diameter, not hailstone embryo diameter.

- Line 256: Would change "since" to "and", as the following phrase agrees with your previous phrase, but does not offer a possible causal factor.

- Lines 2.60-261: Do these uncertainties mean it is possible potential biological aerosol particles were misclassified in your study? If not, what is the reasoning for including this statement here?

- Line 288: Shouldn't the geometric mean diameter be $D_g$ or $d_g$, not $r_g$? r could be too easily confused with radius.

- Fig. 9 is much too small to make out necessary detail. While I appreciate the authors' conscientiousness in ensuring the reader is aware of the geographical locations where the hailstone samples were sourced, I think the responsibility rests with the reader at this point and the maps are no longer necessary. I would split this figure into 2-3 figures to allow points to be made about each individually, as there is a lot of information here. Plus, more detail can be gleaned.

- Section 3.4: Is this section about particle concentrations from the embryos, the shells, or both? Are there concentration size distributions of these particle types for the air at large, or in emissions from specific cities, that these distributions could be compared to? What does having these equations accomplish?

- Lines 325-326: I'm not sure why this statement couldn't be gleaned from Fig. 8 alone, without needing to fit to Eq. 12.

Grammatical/Typographical corrections:
There are quite a few minor grammatical errors throughout, things like "a" or "the" missing before words, misplaced commas, or subject/verb tense agreement. These don't obscure the science being presented, but I recommend the authors ask for proofreading help from a source with professional proficiency in English. I've included some examples from the first couple pages below.
- Line 13: "to little regard paid to"
- Line 16: "A total of 289,461..."
- line 17: comma after bioprotein
- Line 17: vary → varies, in → among
- Line 18: "were performed as" → "were found to follow"
- Throughout: need a space between last letter of a word and the first parenthesis of a citation
- Line 27: "that leads" → "leading"
- Line 27: Add "the" before "number concentration"
- Line 235: "graupels" → "graupel particles"

---

## Author Comment (AC1)

**egusphere-2023-290**
**Reply #1:**

Insoluble aerosol particles are the main source of ice nucleating particles (INPs) in clouds, but their physical and chemical properties have still not been well determined. This manuscript presents analysis of insoluble particles in 12 hailstones collected from 8 hailstorms occurred in China between 2016 and 2021, by scanning electron microscopy (SEM) and energy dispersive X-ray spectrometry (EDX). The insoluble particles were grouped into three species by self-organized maps (SOM) and random forest method. The size distribution of the insoluble particles in embryos and different shells of sliced hailstones was analyzed and was fitted with logarithmic normal distributions. The subject is scientifically interesting and is well within the scope of the journal. But some parts of the manuscript are not well presented or not clear. I think the presenting quality should be substantially improved before it can be accepted for publication in ACP. Some more specific comments are as follows.

Thank you for your review. We have carefully considered every comment and incorporated all of the suggestions into the revised manuscript. The following texts are our point-to-point response.

**Specific comments**

1. Although a large number of insoluble particles were found in each of the hailstone samples, it is not sure whether and how many those insoluble particles have ever served as ice nuclei during the formation of the embryos and different shells of hailstones, since many of them might be captured during the formation, growth and falling out of the hailstones.

   Reply: Yes. We agree that we may not determine how many these insoluble particles have ever served as ice nuclei. This work is the very first step to answer this question. We collaborated with a PH. D student from another group, who found a positive correlation between insoluble particles we measured and immersion freezing nucleation particles in hailstones. The result is currently under review in Atmospheric Research. This finding is consistent with a parameterization for freezing nucleation that was developed based on the concentration of ice-nucleating particles and aerosols larger than 0.5 μm (DeMott et al., 2010).

   Our subsequent objective is to quantify the number concentration of immersion ice-nucleating particles and to determine the type of ice-nucleating particles in each shell in drop-freezing experiment. Then, we will establish a relationship between immersion ice-nucleating particles and size distribution of insoluble particles.

2. Some sentences are not clearly presented to the readers, such as "… simulation due to little regarding species and number concentration of heterogeneous ice-nucleating particles" in the abstract. I suggest the whole text be checked with the help of an English editor.

   Reply: We have made some changes to the manuscript in an effort to improve it. These changes do not affect the content or structure of the paper, and we have not listed all of them here. We

would like to express our gratitude for the reviewers diligent work and h ope that these corrections will be satisfactory.

3. Line 18-19: "Further, classic size distribution modes of organics and dust were performed as logarithmic normal distributions": not clear.

    Reply: The text has been revised as "classic size distribution of organics and dust followed logarithmic normal distributions"

4. Line 22-23: "Insoluble particles, acting as main heterogeneous ice-nucleating particles in the atmosphere, may indirectly impact precipitation formation": why indirectly?

    Reply: "Indirectly" was removed.

5. Line 26-28: "Only few models calculate the number concentration of ice-nucleating particles in clouds, that leads to a misestimation about number concentration of ice particles and large errors in simulation": not clear to me.

    Reply: The text has been revised as "Few models used the freezing parameterization, which establishes a connection between the number concentration of ice-nucleating particles and the number concentration of ice crystals explicitly. The absence of any description regarding the physical properties of ice-nucleating particles in models can result in an incorrect estimation of ice crystals and lead to significant bias in simulations".

6. Line 31-32: "Measurements of the number concentration and species of ice-nucleating particles, mainly insoluble particles, were conducted by an airborne equipment or laboratory instrument with air parcels": not clear to me.

    Reply: There are two situation for sampling ice-nucleating particles.

    The first involves airborne instruments, such as the continuous flow thermal gradient diffusion chamber (DeMott et al., 2010). This technique collects air during flight, activates ice-nucleating particles, and counts ice crystals. It can measure the number concentration of insoluble particles in several freezing modes (up to four modes) by changing the temperature and supersaturation in the chamber.

    The second is in laboratory where scientists measure the number concentration of ice-nucleating particles in air parcels. The air sample was sampled by aircraft in the air (e.g., Winter Icing in Storms Project 1994) or inlet at high altitude (e.g., Winter Icing in Storms Project 2000). Additionally, certain species of particles aerosols are tested in laboratories to determine their freezing efficiency.

    Lines 33-36 have been expanded to include additional details as: "There are two ways to sample ice-nucleating particles: The first involves an airborne instrument, named continuous

flow thermal gradient diffusion chamber. The second is done in the laboratory, where scientists conduct freezing experiments".

7. Line 34-35: "Most field projects sampled air parcels in anvils of convective clouds, cirrus and winter mixed-phase stratiform clouds, keeping airborne equipment in good working condition": I am not sure what this sentence want to tell.

Reply: As mentioned in response to point 6, airborne continuous flow thermal gradient diffusion chamber is used to count ice-nucleating particles in sampled air while the aircraft is flying. These detections take place in cloud-free regions (Winter Icing in Storms Project 1994 and Mixed-Phase Arctic Cloud Experiment), inside plumes (PACDEX), within the base of stratiform clouds (PACDEX), and through cirrus clouds (PACDEX).

Bad weather events can hinder flights during field projects. For example, turbulence disrupted communication systems during the 6th research flight of Alliance Icing Research Study-2. Pilots on the 11th research flight were worried about that cloud top transit could create problems if long-term supercooled conditions were experienced.

These weather-related events were recorded in flight logs, but not all field studies share their flight reports with open access. Reports and articles indicate that no flight can sample air parcels through cores in convection, especially deep convection in severe storms.

We have removed "keeping airborne equipment in good working condition" and the word "most". The Lines 33-41 has been revised for clarity as "T There are two ways to sample ice-nucleating particles: The first involves an airborne instrument, named continuous flow thermal gradient diffusion chamber(DeMott et al., 2010; Prenni et al., 2009; Rogers et al., 2001). The second is done in the laboratory, where scientists conduct freezing experiments (Hoose and Möhler, 2012). In most cases, it is necessary for aircraft to collect air parcels for measurement of the physical properties of ice-nucleating particles in the air. However, former field projects sampled air parcels in anvils of convective clouds, cirrus and winter mixed-phase stratiform clouds. No flight report or article has reported that they sampled air parcels through cores in deep convection. This phenomenon is consistent with consideration for flight security. Thus, current observation is insufficient for describing the whole convective cloud, especially the deep convection in severe storms."

8. Line 39-41: The logics of paragraph is incorrect.

Reply: Thank you. We found the logical mistake and modified the sentence. The text has been revised as "Hailstones, as a product of deep convective clouds, serves as a carrier of information within these clouds. Recently, analysis revealed large diversity in number concentration of soluble ions among hailstones from different hailstorms (Li et al., 2018). Further, the detection of soluble ions along with isotopic analysis of a huge hailstone revealed an up-and-down hailstone growth trajectory, which demonstrated that the different shells were formed at different heights (Li et al., 2020)."

9. In Table 1, why only one value is provided for the two samples from Guyuan City?

Reply: We have listed the particles number of each hailstone as your suggestion and revised footnote of Table 1.

10. In Line 169, Formula (2): why $N_{used} = N_{filter}$?

Reply: It is an assumption. The number of insoluble particles present in the consumed diluting solution $(N_{used})$ range from $10^5$ to $10^6$. It is expected as a true value. As mentioned in lines 100-101, we repeated rinsing for 5 times to ensure particles adhered to the membrane as much as possible, so that $N_{used}$ was assumed to be equal to the number of insoluble particles found on the filter membrane $(N_{filter})$.

Additional details regarding this process have been included in lines 226-228 of our manuscript as "Assuming the rinsing operation ensures all insoluble particles in the shell were on the membrane, the number of insoluble particles in the consumed solution $(N_{used})$ is equal to the number of insoluble particles counted on the membrane $(N_{filter})$."

11. Line 173, formula (3): The inversion form of this formula might be easier to be understood.

Reply: Thank you for your comment. The formula has been revised to:

$$\frac{S_{images}}{S_{filter}} = \frac{N_{count}}{N_{filter}} \qquad (R1-1)$$

12. Line 182: How formula (5) is derived?

Reply: The Formula (4) is:

$$n_{liquid} = \frac{1}{V_{liquid}} \cdot \frac{S_{filter}}{S_{images}} \cdot \frac{V_{diluted}}{V_{used}} \cdot N_{count} \qquad (R1-2)$$

Take the logarithm on both sides:

$$\ln n_{liquid} = -\ln V_{liquid} + \ln S_{filter} - \ln S_{images} + \ln V_{diluted} - \ln V_{used} + \ln N_{count}$$

Next, differentiate the equation:

$$\frac{dn_{liquid}}{n_{liquid}} = -\frac{dV_{liquid}}{V_{liquid}} + \frac{dS_{filter}}{S_{filter}} - \frac{dS_{images}}{S_{images}} + \frac{dV_{diluted}}{V_{diluted}} - \frac{dV_{used}}{V_{used}} + \frac{dN_{count}}{N_{count}}$$

As,

$$dS_{filter} = dS_{images} = 0$$

Now, we get the formula (5):

$$dn_{liquid} = n_{liquid} \cdot \left( \frac{dV_{liquid}}{V_{liquid}} + \frac{dV_{diluted}}{V_{diluted}} + \frac{dV_{used}}{V_{used}} + \frac{dN_{count}}{N_{count}} \right) \qquad (R1-3)$$

We also add more description into the manuscript.

13. Line 227-228: "Each number concentration at diameter D total number concentration of insoluble particles with diameter ranging from D−1 μm to D+0.1 μm", not clear to me;

Reply: The bin width is 0.2 μm in Fig. 7 and Fig. 10, and 2 μm in Fig. 8 and Fig. 9.

14. Line 230-231: "Blue and gray bars show the standard deviation of insoluble particles from seven hailstorms and one hailstorm, respectively": not clear.

Reply: Blue bars show the standard deviation of the number concentration of insoluble particles in hailstones (BJ1, BJ2, BS, FS,GY1, GY2, YT and GYA) from eight different hailstorms. On the other hand, gray bars represent the standard deviation of the number concentration of insoluble particles in five hailstones (BJ2 to BJ6) collected from the same hailstorm. The caption has been revised.

[Figure]

Fig. 7: Size distribution of organics, dust, and bioprotein aerosols of insoluble particles in 12 hailstones. The colored dots represent data from 7 hailstones BJ1, BS, FS,GY1, GY2, YT and GYA which were from different hailstorms. The black and gray dots correspond to data from hailstones (BJ2 to BJ6) that were from the same hailstorm occurring in Beijing on June 30, 2021. The blue and gray bars indicate the standard deviation of number concentration of insoluble particles from 8 hailstones (BJ1, BJ2, BS, FS,GY1, GY2, YT and GYA) from 8 cases and 5 hailstones (BJ2 to BJ6) from one case, respectively. Abbreviations (corresponding to Table 1): BJ - BeiJing; BS - BaiSe; FS - FuShun; GY - GuYuan; GYA - GuiYAng; YT - YanTai.

15. Line 236-237: "These initial ice particles are likely formed by insoluble particles where heterogeneous nucleation processes": incomplete sentence.

    Reply: We completed this sentence. Text has been revised as "These initial ice particles are formed through nucleation of insoluble particles where heterogeneous nucleation take place."

16. Line 294 and other places: There should be no unit for logarithmic function.

    Reply: Thank you. We deleted the units of logarithmic terms in the Lines 359-360 and Lines 369-370.

**Technical corrections**

1. Change "for" to "of" after "description" in line 29;
   Text revised.

2. Change "in" to "of" after "suppression" in line 54;
   Text revised.

3. Table captions should be provided on top of the tables;
   Table revised. Please see Table 1 (line 81).

4. Line 116: Change "Ault et al. in 2012 and Kirpes et al. in 2018" to "Ault et al. (2012) and Kirpes et al. (2018)";
   Text revised.

5. Line 117: Remove "Ault et al. 2012; Kirpes et al. 2018";
   Text revised.

6. Line 122: Change "Species of aerosol particles vary regionally" to "Species of aerosol particles vary with sampling location";
   Text revised.

7. Line 166: Formula (1): Change $N_{dilute}$ to $N_{diluted}$;
   Text revised.

8. Line 236: Change "droplet" to "droplets".
   Text revised.

**References:**

DeMott, P. J., Prenni, A. J., Liu, X., Kreidenweis, S. M., Petters, M. D., Twohy, C. H., Richardson, M. S., Eidhammer, T., and Rogers, D. C.: Predicting global atmospheric ice nuclei distributions and their impacts on climate, Proc. Natl. Acad. Sci., 107, 11217–11222, https://doi.org/10.1073/pnas.0910818107, 2010.

Hoose, C., Kristjánsson, J. E., Chen, J.-P., and Hazra, A.: A Classical-Theory-Based Parameterization of Heterogeneous Ice Nucleation by Mineral Dust, Soot, and Biological Particles in a Global Climate Model, J. Atmos. Sci., 67, 2483–2503, https://doi.org/10.1175/2010JAS3425.1, 2010.

Li, X., Zhang, Q., Zhu, T., Li, Z., Lin, J., and Zou, T.: Water-soluble ions in hailstones in northern and southwestern China, Sci. Bull., 63, 1177–1179, https://doi.org/10.1016/j.scib.2018.07.021, 2018.

Li, X., Zhang, Q., Zhou, L., and An, Y.: Chemical composition of a hailstone: evidence for tracking hailstone trajectory in deep convection, Sci. Bull., 65, 1337–1339, https://doi.org/10.1016/j.scib.2020.04.034, 2020.

Prenni, A. J., Demott, P. J., Rogers, D. C., Kreidenweis, S. M., Mcfarquhar, G. M., Zhang, G., and Poellot, M. R.: Ice nuclei characteristics from M-PACE and their relation to ice formation in clouds, Tellus B, 61, 436–448, https://doi.org/10.1111/j.1600-0889.2009.00415.x, 2009.

Rogers, D. C., DeMott, P. J., Kreidenweis, S. M., and Chen, Y.: A Continuous-Flow Diffusion Chamber for Airborne Measurements of Ice Nuclei, J. Atmos. Ocean. Technol., 18, 725–741, https://doi.org/10.1175/1520-0426(2001)018<0725:ACFDCF>2.0.CO;2, 2001.

---

## Author Comment (AC2)

**egusphere-2023-290**
**Reply #2:**

This article provides some of the first observations of insoluble particle concentrations within hailstones collected at the ground, as well as the opportunity for embryo analysis. My primary concern is that while this observational dataset is unique and worthwhile, the authors do not use it to draw any conclusions about physical processes related to hail growth. It isn't clear what the purpose of these observations is. The description of the classification and clustering technique is also confusing. Once these issues are addressed I am comfortable with publication.

Thank you for your positive evaluation. We have made effort to clarify every comment and implemented all the suggestions in the revised manuscript. The following texts are our point-to-point response. Two additional references have been included in the main text to provide a more detailed description of SOMs algorithms and uncertainties calculation:

Kohonen, T.: The self-organizing map, Proc. IEEE, 78, 1464–1480, https://doi.org/10.1109/5.58325, 1990.

Taylor, J. R.: An Introduction to Error Analysis, Second edi., University Science Books, 330 pp., ISBN 093570275X, 1997.

**Major Comments:**

1. The authors have clearly spent a lot of effort to gather these observations, and they are some of the first studies of insoluble particles in hailstones. However, I would like to see some conclusion about the physical processes that these observations can now help us to understand. What, physically, can we now conclude about hailstone growth, or insoluble particles in hailstones, that we couldn't before? As it stands, the article is simply a reporting of what the authors found. What do these observations mean?

Reply:

The general goal of our work is to study the physical processes related to hailstone growth.

Freezing parameterizations are developed by concentration measurement of ice-nucleating particles in the air, under clear or slightly cloudy conditions, or near the surface. Please refer to the Reply 1, point 6. On the one hand, the freezing parameterization in models considers little about the impact of aerosol particle concentration and species. On the other hand, the ice-nucleating parameterization describes the freezing process of dry particles, which is not suitable to describe the hailstone growth by freezing supercooled droplets. These two factors contribute to the limitations of the current freezing parameterizations in accurately describing the hailstone growth.

A two-order-of-magnitude change in aerosol concentration in the air leads to at least a one-order-of-magnitude change in the concentration of ice-nucleating particles (DeMott et al., 2010). Our experiments revealed that the insoluble particle concentrations in different deep convections vary by one to two orders of magnitude. Therefore, when simulating a specific deep convection case, it is necessary to consider the variations in aerosol concentration within clouds. Further, our research identified the species and number of insoluble particles within deep convective clouds. It provided a new approach to establish the freezing parameterization, which would consider aerosol

concentration, aerosol species, and hailstone growth mechanism in the hailstorms.

What information does the knowledge about how the particle distributions change among embryos/shells provide about physical processes involved in hailstone growth? For example:
• Which hailstones had size distributions that changed significantly between one shell to the next? Why might this occur?
Reply:
    We guess the "size distributions" you mentioned is "the size distributions of insoluble particles".

[Figure]

Reply2. Figure. 1 The ratio of the number concentration of insoluble particles vary in shells in hailstones. Blue, orange and purple are used to indicate dust, organics, and bioprotein aerosols, respectively. Abbreviations (corresponding to Table. 1): BJ - BeiJing; BS - BaiSe; FS - FuShun; GY - GuYuan; GYA - GuiYAng; YT - YanTai.

Among two-shell hailstones, GY2 and GYA changed a lot between the embryo and the crust. This suggests that the graupel may form at high altitudes within the cloud, where fewer particles existed in supercooled water droplets. Subsequently, the graupel descended and collected water droplets with more particles at lower levels. For BS, the formation of the embryo and the shell B may be contrary to the growing process of GY2 and GYA.

• Do shell B and embryo distributions usually look the same or different? Shell C and B? Do size distribution/particle type vary more among shells, or among hailstones? Why?

Reply:

Yes, insoluble particles in different shell (shell B and embryo, shell C and shell B, and Shell C and crust) looks similarity in size distribution. Compared to it, the particle distributions vary more among cases. The reason is the updrafts can transport the pollution in local boundary layer into the clouds as hailstorms are severe convections.

• Do you have a sense if these hailstones might have taken similar paths through the storms that generated them?

Reply:

According to our unpublished work, shells in hailstones formed at different heights within hailstorms. It indicates that the paths of different hailstones show large variance.

• Were the storms that generated these hailstones all of similar convective mode, or different? Would you expect storms with stronger updrafts to transport more particles, changing the insoluble particle size distribution? Why or why not?

Reply:

The storms were single cells and squall lines. We currently thought the convective modes were different.

A former study show significant correlation between soluble ions in hailstone and surface aerosols, indicating the aerosols in deep convection may come from local area (Li et al., 2018). A stronger updraft can transport more insoluble particles into clouds. But too strong updraft can eject hail embryos from the growth region and hinder their mature growth (Fan et al., 2022). Thus, a proper stronger updraft may support hailstones to pass through a larger region within the convection and then to collect more particles from the surface, resulting in a more significant variation in particle size concentration.

2. The description of the classification and clustering techniques in Section 2.2 is very difficult to understand, making the subsequent results in the rest of the paper unnecessarily hard to follow.

• Fig 2: This figure is hard to follow. Id label each step with a single action (e.g., polishing, slicing, shell extraction, etc.), and then list and describe each action by name in the caption (or in the text itself) Numbering the steps and boxes would also help. That way it is clear which stage of the process corresponds to which box in the figure. Names of equipment can go below each image, in smaller font perhaps.

Reply:

Thank you for kind comments. Fig. 2 has been modified as below:

[Figure]

Fig. 2: Schematic diagram illustrating the experimental framework. [1-2] The surface of each hailstone was polished to remove any attached grass or soil. [3] Subsequently, the hailstones were sliced into cross-sections along the major axis, corresponding to the size of the hailstone embryo. [4-7] After photographing the hailstone cross-sections, they were further subdivided into shells using heated Fe-Cr alloy wire at an air temperature below −8°C. The shells were distinguished based on their natural transparency or opacity. [8] The solution of melting shell samples was then passed through a filter membrane to isolate the insoluble particles. [9] Each shell sample underwent analysis using scanning electron microscopy and energy-dispersive X-ray spectrometry to determine the elemental weight ratios of the insoluble particles within approximately 4 hours. [11] Finally, the elemental weight ratio information of hailstones was obtained.

• Lines 126- 127: This step should be included as a step in Fig. 3, showing what M1 - M100 are and how they are generated.

Reply:

Fig. 3 has been modified as below:

[Figure]

Fig. 3: Schematic diagram illustrating the methodological framework used for particle identification in this study. A total of 100 matrices $M_i$, with $i$ ranging from 1 to 100, were utilized in self-organized maps clustering analyses, each containing 81,888 unidentified particles with 19 elemental features (N, Na, Mg, Al, Si, P, S, Cl, K, Ca, Ti, Cr, Mn, Fe, Ni, Cu, Br, Ba, and Pb). The centroid matrix $C_{k,i,j}$ represents the clustering results obtained through the self-organized maps method with a given cluster number $k$. The self-organized maps operation with the same $k$ was repeated 100 times to ensure result robustness, where $j$ denotes the number of repetitions ranging from 1 to 100. Four indexes, Silhouette index (Sil), Calinski–Harabasz index (CH), modified Hartigan index (Hart), and Davies–Bouldin index (DB), were employed to determine the optimal parameters $k$, $i$, and $j$. The matrix $M_i$ containing identified 81,888 particles was randomly divided into a training set (80%) and a test set (20%) for random forest classification. The 10-fold cross-validation was utilized to determine the best tree. Abbreviations (corresponding to Table. 1): BJ - BeiJing; BS - BaiSe; FS - FuShun; GY - GuYuan; GYA - GuiYAng; YT - YanTai.

• Lines 128- 137: The SOMs description could be clearer. What is being produced by this process? Identification of particle species? How is that determined from a given centroid matrix? Are the number of outcome clusters predefined (a perhaps that is k?) If so, how are the numbers k-2 through 10 selected?

Reply:

Self-Organizing Maps (SOMs) are a type of artificial neural network that belongs to the category of competitive learning algorithms (Kohonen, 1990). This network emulates the self-organizing feature mapping function observed in the neural network of the human brain. SOMs are commonly used for performing dimensionality reduction algorithms, enabling the representation of high-dimensional data in a lower-dimensional structure while preserving the original topology. A basic self-organizing maps structure consists of an input layer, weight vectors, and an output layer. These weight vectors represent the topological structure of the neurons in the output layer.

Identification of particle species and matrix $C_{k,i,j}$ are produced by SOMs. The matrix $C_{k,i,j}$

describe the statement of the neurons, which provides an average representation of the cluster, i.e., the centroid of the cluster.

The value of k is predetermined.
In supervised methods, the number of outcome clusters can be determined based on labeled data. However, unsupervised methods utilize unlabeled data for training, which makes it uncertain how many types of input data exist. In unsupervised clustering, it is common to determine the number of clusters (k) and subsequently evaluate the performance of clustering. SOMs belong to unsupervised methods, so that the value of k needs to be predefined.

The value of k can vary between 1 (all samples belong to the same cluster) and 81888 (each particle belonging to an individual cluster). Classification is meaningless when k equals to 1. Thus, we initialized k with a value of 2. The Fig. 4 demonstrated that the optimal classification performance was achieved when k was 6, and each centroid in $C_{k,i,j}$ presented interpretability. The classification performance gradually got worse from $k = 6$ to $k = 10$. For saving computational resources, we limited the value of k up to 10.

[Figure]

Fig. 4: Evaluation of self-organized maps clustering results. The clustering results of self-organized maps were evaluated using (a) Silhouette index, (b) Davies–Bouldin index, (c) Calinski–Harabasz index, and (d) Hartigan index. The self-organized maps operation was repeated 100 times to ensure result robustness. The solid lines and shading represent the average and spread of 100 repetitions, respectively.

• Lines 133-134: Does the "same neuronal network setting" mean k is the same, or k, i, and j are all the same? For that matter, what does "neuronal network setting" mean?
Reply:
The configuration of the neuronal network includes the dimensions of the output layer, the number of training steps for the initial coverage of the input space, the initial neighborhood size,

and the type of topology function. The "same neuronal network setting" refers to the same configuration of the neuronal network, including the same dimensions of the output layer (k neurons) and the same input data ($M_i$). The j changed.

• What are "particle sample replicates"?

Reply:

It refers to the times of extracting 81,888 insoluble particles to compose the $M_i$. The text has been revised.

• Lines 134-136: What information does each of these indices provide? Most importantly, how do you determine accuracy with this method?

Reply:

Unsupervised methods utilize unlabeled data for training. Therefore, determining the optimal number of clusters in unsupervised classification is a challenge. To address this, statisticians have developed indices that evaluate the performance of SOMs and assist in selecting the optimal number of clusters (k).

We provide a simple example demonstrating how to calculate the Silhouette index. The Silhouette index was defined as (Rousseeuw, 1987):

$$Sil = \sum_n \frac{b(n) - a(n)}{max a(n), b(n)} / N \qquad (R2 - 1)$$

For example:

$$a(n) = \frac{a_1 + a_2 + a_3 + a_4}{4} \qquad (R2 - 2)$$

$$b(n) = \frac{b_1 + b_2 + b_3}{3} \qquad (R2 - 3)$$

For each particle $Particle_n$ (where n ranges from 1 to 81,888) in matrix $M_i$, we define two variables: $a(n)$, the average Euclidean distance from $Particle_n$ to all other particles within the same cluster (cluster A); $b(n)$, the average Euclidean distance from $Particle_n$ to all particles within the nearest cluster (cluster B). Additionally, the Silhouette index ranges from -1 to 1, where a value closer to 1 indicates that each particle is well-matched to its own cluster, while a value closer to -1 suggests that some particles would be better suited in a different cluster.

[Figure]

Reply2. Figure. 2 Clusters illustration

The Calinski–Harabasz index (Calinski and Harabasz, 1974) is defined as:

$$CH = \frac{\sum_{g=1}^{k} n_g (C_g - C)^2}{\sum_{g=1}^{k} \sum_{x \in C_g} (x_g - C_g)^2} \times \frac{N - k}{k - 1} \qquad (R2 - 4)$$

k : the number of clusters.

C : the centroid of all data

N : the number of observations in data

$C_g$ : the centroid of cluster g

$n_g$ : the number of observations in cluster g

$x_g$ : the observation of cluster g.

Where $\sum_{g=1}^{k} n_g (C_g - C)^2$ represents the sum of squared distances between cluster centroids $C_g$ and the overall centroid C, $\sum_{g=1}^{k} \sum_{x \in C_g} (x_g - C_g)^2$ represents the sum of squared distances within each cluster. A higher value of the CH index for given k indicates better clustering performance, as it suggests a larger ratio between inter-cluster dispersion and intra-cluster dispersion.

The Davies–Bouldin index (Davies and Bouldin, 1979) is defined as:

$$DB = \frac{1}{k} \sum_{\substack{g,f=1}}^{k} \max_{p \neq q} \frac{s_g + s_f}{\sqrt{(C_g - C_f)^2}} \qquad (R2 - 5)$$

$$s_g = \sqrt{\frac{(x_g - C_g)^2}{n_g}}, s_f = \sqrt{\frac{(x_f - C_f)^2}{n_f}} \qquad (R2 - 6)$$

$x_f$ : the observation of cluster f

$C_f$ : the centroid of cluster f

$n_f$ : the number of observations in cluster f

DB index measures the maximum ratio of intra-cluster dispersion to inter-cluster dispersion. A lower DB index indicates better clustering performance.

The Silhouette index, Calinski–Harabasz index, and Davies–Bouldin index, assess the similarity between a particle and others within the same cluster, as well as the dissimilarity across different clusters for given k. Generally, the effectiveness of clustering fluctuates with k. Is it beneficial to increase the value of k? The Hartigan index, on the other hand, quantifies the improvement of clustering efficiency from k to k + 1.

Hartigan index (Sibson and Hartigan, 1976) is defined as:

$$H(k) = (N - k - 1)\left[\frac{err(k)}{err(k+1)} - 1\right] \qquad (R2-7)$$

$$err(k) = \sum_{g=1}^{k}\sum_{x_g \in C_g}(x_g - C_g)^2 \qquad (R2-8)$$

The H(k) requires clustering results with k from 2 to 11 in order to obtain H(2), H(3), …, H(10). Clustering particles into 11 clusters needs an additional 100 iterations of SOMs for each $M_1$ to $M_{100}$. Additionally, we observed that the SOMs did not perform well in Silhouette index (Sil), Calinski–Harabasz index (CH), and Davies–Bouldin index (DB) when k = 2. Therefore, we made modifications to the Hartigan index and introduced Hart(k):

$$Hart(k) = [N - (k-1) - 1]\left[\frac{err(k-1)}{err(k)} - 1\right], k = 2\sim10 \qquad (R2-9)$$

$$err(k) = \sum_{g=1}^{k}\sum_{x \in C_g}(x_g - C_g)^2, k \geq 2 \qquad (R2-10)$$

When k = 1, it indicates that all particles are belong to one cluster.

$$err(1) = \sum_{n=1}^{N}(x_n - C)^2 \qquad (R2-11)$$

C : the centroid of all data

$x_n$ : the observation of data

The err(k) describes intra-cluster dispersion. In clustering with a specific value of k, our objective is to have particles tightly grouped together in feature space while ensuring that the centroids exhibit a significant dispersion compared to k - 1. A higher value of Hart(k) for a given k indicates improved clustering performance.

Technologically, we employ a repetition approach to cluster particles and evaluate the results of SOMs for 100 iterations (as indicated by the shading in Fig. 4) to ensure the robustness of SOMs outcomes for a given $M_i$ and k. Rescale Sil(k, i, j), CH(k, i, j), DB(k, i, j) and Hart(k, i, j) by max normalization, then the best input $M_i$ and the iterate times j satisfied:

$$maximum\{Sil(k, i, j) + CH(k, i, j) + Hart(k, i, j) - DB(k, i, j)\} \qquad (R2-12)$$

The indices serve as indicators for determining the appropriate value of k. However, it is crucial

to consider the meaning of centroids in determining $k$. In our work, when $k = 6$, it satisfied the requirements for robustness, great statistical performance in clustering, and interpretable clustering outputs.

• Lines 144- 149, Fig. 5: Unfortunately, I cant follow this description at all. What information does a centroid matrix have in it? Are carbon and oxygen included in the classification matrix, and as SOM inputs, or not? Lines 118- 121 seem to indicate no.
Reply:

The centroid matrix $C_{k,i,j}$ approximately represents k centroids of the clustering results.

A centroid matrix contains the elemental weight ratios of 19 elements in k neurons in the output layer except carbon and oxygen.

Neither carbon nor oxygen were included in the classification matrix or as SOMs inputs. Nonetheless, carbon and oxygen ratios can provide valuable information and aid in species identification. For instance, the first type of particles exhibits a low content of 19 elements in their elemental weight ratios. It is insufficient for determining the composition of this substance. Consequently, we also consider the mean carbon and oxygen ratios of particles of this type to aid in identification.

• Lines 144 -159: Please explicitly state what is being used as the "truth" dataset for the random forest. I had thought it would be the centroid matrix (line 144), but instead I think it is the classification outcome of the centroid matrix (Fig. 6)?
Reply:

The "truth" refers to labeled particles, which is the classification outcome from the SOMs. The centroid matrix is not used as an input in the random forest algorithm. It just helped us in understanding substance of each cluster.

3. The number of studies of observed hailstone embryos in the literature less than 30 years old is almost zero, so it seems a real missed opportunity not to offer some details about the embryos collected here. These were all graupel embryos, not frozen drops? How do the characteristics of these embryos and hailstorms correspond to the hailstone embryo research of Knight (1981, J. Appl. Meteorology and Climatology)? How big were each of the embryos? Are you able to estimate their density? Differently sized embryos would also have to have impacts on their insoluble particle makeup, I would think. Do you find that to be the case?
Reply:

All the embryos analyzed in our study were graupels.

Knight established the correlation between cloud base temperature and the occurrence likelihood of types of hail embryos by statistically analyzing thousands of hailstone embryos. With the help of volunteers, we collected hailstones, but the amount was extremely limited. As a result, we were unable to reach the same order of magnitude as Knights observation. Nonetheless, we attempt to show a comparison of hail embryo types and cloud base temperature in our work with the findings from Knights research.

Soundings in China are twice daily, at 8:00 AM and 8:00 PM, Beijing local time. Assuming the change in the upper sounding curve is negligible and can be disregarded. The lifted condensation level (LCL) is calculated using soundings conducted both before and closer to the occurrence of hail. Temperature and dew point temperature in soundings within a height range from the surface up to 1 km above the surface are adjusted using automatic weather station data. To verify the method, the LCL derived from ERA5 reanalysis data matches the results obtained from sounding except BJ1

Reply2. Table. 1 The date, time, latitude and longitude where the hailstone were collected. The type of hail embryos, sample abbreviation and LCL were listed.

| Case | Date & Beijing Time | Latitude & Longitude | Number of hailstones | Location & Sample abbreviation | LCL from sounding (℃) | LCL from ERA5 (℃) |
|---|---|---|---|---|---|---|
| 1 | 19 June 2018, 18:30 | 41.82° N, 123.85° E | 1 | FuShun (FS) | 17.28 | 16.18 |
| 2 | 10 June 2016, 15:00 | 40.00° N, 116.32° E | 1 | BeiJing (BJ1) | 15.08 | 15.20 |
| 3 | 30 June 2021, 20:18 | 39.95° N, 116.30° E | 5 | BeiJing (BJ2 – BJ6) | 17.16 | - 4.87 |
| 4 | 01 Oct 2021, 14:02 | 37.49° N, 121.44° E | 1 | YanTai (YT) | 19.04 | 19.38 |
| 5 | 25 Aug 2020, 18:00 | 35.53° N, 106.32° E | 1 | GuYuan (GY1) | 4.94 | 4.20 |
| 6 | 26 Aug 2020, 16:00 | 35.58° N, 105.93° E | 1 | GuYuan (GY2) | 4.56 | 3.69 |
| 7 | 14 Apr 2016, 20:00 | 26.60° N, 106.72° E | 1 | GuiYAng (GYA) | 17.11 | 17.21 |
| 8 | 09 May 2016, 18:51 | 23.90° N, 106.60° E | 1 | BaiSe (BS) | 20.83 | 20.85 |

Our result is not consistent with Knights. Whether the cloud bases are cool or warm, hail embryos are graupels. Two reasons contribute to this discrepancy. Firstly, the number of analyzed samples is relatively small due to the limited quantity of samples collected with the assistance of volunteers, resulting in a sampling bias. Secondly, there is notable inconsistency in the cloud base types between our cases and Knights study. While Knights study had 4 out of 5 cases with cool cloud bases, our research had 6 out of 8 cases with warm cloud bases. When comparing the hail embryo types corresponding to cool cloud bases, the inconsistency between our conclusion and Knights findings reduced .

[Figure]

Reply2. Figure. 3 Frequency of embryo types plotted against the average cloud-base temperature. Adapted from Knight (1981). Hail embryos from my experiment are indicated by gray circles.

We noticed your interest in the size and weight of hailstone embryos. Unfortunately, it is not the purpose of this study and we did not measure or weigh it. Currently, we cannot summarize the relationship between embryo size and particle distribution. We appreciate your suggestion, and will add measurement of size, weight and solid volume of hailstone embryos in future work.

**Minor Comments:**

• Lines 170- 173: I think I get your meaning here, but it should be clearer. Do you mean that because you assume that the random subsample of the filter is representative of the entire filter, Ncount is determined by multiplying the observed Nfilter by the area ratio between the whole filter and the obsered image (Simages/Sfilter)? If so, I would explain it like that.
Reply:
Yes, assuming a uniform distribution of insoluble particles on the filter membrane, a software randomly capture electron microscope photos of the membrane and count the visible insoluble particles in those images. The total number of insoluble particles on the filter membrane ($N_{filter}$) is

calculated by multiplying the number of visible insoluble particles counted in the images ($N_{count}$) by the ratio of the areas between the entire filter membrane and the observed images $\left(\frac{S_{filter}}{S_{images}}\right)$. This assumption allows us to estimate the total number of insoluble particles on the filter membrane based on subsampling areas.

• Lines 174-177: Move these sentences to the start of the subsection immediately following (1). Also, adding a sentence after each equation explaining the physical meaning of it would be helpful In the reader. E.g., The number of insolvable particles in the melted shell (Nliquid) can be found by multiplying their number concentration (nliquid) by the volume of the melted shell (Vliquid); this total particle number does not change when the solution is diluted (Ndilute).
Reply:

    Thanks for your comment. We add sentences after each equation to help readers understand the equations in the context.

• Line 180: How is Ncount determined? I thought only Nfilter could be observed.
Reply:

    $N_{count}$ corresponds to the number of counted insoluble particles in electron microscope images by a software. $N_{filter}$ is derived from $N_{count}$ using Eq. (8). This equation enables us to estimate the total number of particles present on the filter membrane ($N_{filter}$) by counting insoluble particles on the relative areas ($N_{count}$) involved in the observation process.

• Lines 185- 188: How are these equations determined?
Reply:

    We assumed that the uncertainties of the equation terms are independent and random. These equations were derived from uncertainties calculation (Equation 3.18 in Taylor, 1997). To avoid any misunderstanding, we replaced the inequality symbol "<=" in the equation into an equal sign "=" and deleted the Equation 8 of the unrevised context.

• Line 196: What data? Particle number concentrations (nliquid)? Binned by particle diameter size?
Reply:

    The data corresponds to particle number concentrations per unit volume of liquid water. The concentrations were grouped into bins based on particle diameter size. In Fig. 7 and Fig. 10, the bin interval is  0.2 µm, while in Fig. 8 and Fig. 9, it is  2 µm.

• Lines 200 -203: Why was a log-normal distribution chosen? What are rg and Tg? Why this form of the distribution? What does line 202-203 mean physically?
Reply:

    Log-normal distribution is widely used in the description of aerosols (Lamb and Verlinde, 2011). The size distribution of insoluble particles exhibits a normal distribution shape when plotted on a logarithmic scale along the x-axis (Fig. 10). It is important to note that normal distributions can allow negative independent variables, whereas log-normal distributions allow only positive independent variables. The particle diameter size values are always positive. Therefore, we utilized the log-normal distribution for our analysis

We noticed there is a typo where $r_g$ should be replaced with $D_g$. The Eq. (15) has been corrected accordingly. $T_g$ was not mentioned. Did you mean $\sigma_g$?

$D_g$ represents the geometric mean diameter, while $\sigma_g$ represents the geometric standard deviation. The geometric mean diameter serves as the central value in a log-scale distribution, while the geometric standard deviation quantifies the spread of a log-normal distribution. Unlike the arithmetic standard deviation, which measures the differences between data points and the mean value, the geometric standard deviation considers the ratios between data points and the geometric mean.

Lines 202-203 explained the reason of removing the data exhibiting a flat tail caused by conversion during the fitting process. If particles are detected within a bin, the $N_{count}$ of that bin is at least 1. The $n_{liquid}$ within this bin is at least equal to the conversion coefficient as stated in Eq. (9). Bins with $N_{count} = 1$, corresponding to the appearance of a flat-tailed curve. This flat tail is not conducive when fitted a log-normal distribution. Thus, during the fitting process, we excluded the bins with sizes larger than the first size where binned $N_{count}$ reaches 1.

• Lines 216- 225, Fig. 7: Given the log scale and the small y-axes of Fig. 7, it is difficult to see the differences between any hailstones, let alone among specific storms. I recommend shifting the standard deviation results to a new figure. I would also be curious to see what the standard deviation values are across all storms but excluding the GY1 and GY2 hailstones. I am curious how much of the increased standard deviation for all 7 hailstorms in sum is due to those 2 storms. Once those two storms are removed, that should have an impact on the conclusions in Lines 216- 225. Possibly, using just one Beijing hailstone is not representative.
Reply:

For a convenient comparison of the dispersion of number concentration among various species and at the same particle diameters, we did not separate standard deviation results to a new figure.

We agree with you that GY1 and GY2 are major contributors to the standard deviation. This might result from the insufficient amount of samples and our future studies will aim to increase the sample size to strengthen this conclusion. Additionally, the standard deviation is widely recognized as a statistical measure for evaluating data dispersion. We utilize standard deviation as a quantitative indicator of data dispersion. Discussing the dispersion of data after excluding GY1 and GY2 data would be inappropriate.

• Lines 237 - 238: Thats a lot o f "possiblies". One could just as easily argue the insoluble particles were contributed by the riming supercooled water acting to form the embryo.
Reply:

The text has been revised.

• Lines 248-251: I'm having trouble following these sentences. Why would industrial coal burning result in an increased number of organic aerosols specifically 10 microns in diameter?
Reply:

It was our speculation on the origin of coarse organics based on several literatures. We have now revised this sentence.

• Lines 252- 253: Rephrase to make it clear "at the same diameter" refers to particle diameter, not hailstone embryo diameter.

Reply:

Text revised. "at the same diameter" means at the same diameter of insoluble particles.

• Line 256: Would change "since" to "and", as the following phrase agrees with your previous phrase, but does not offer a possible causal factor.

Reply:

We agree, "since" has been replaced by "and".

• Lines 260-261: Do these uncertainties mean it is possible potential biological aerosol particles were misclassified in your study? If not, what is the reasoning for including this statement here?

Reply:

Yes, it is possible potential biological aerosol particles were "misclassified" as organics. The reason is different classification methods.

Aerosols can be classified based on aerosol size, generation, sources, chemical composition, morphology and optical properties, leading to diverse category. Classification by source, aerosols can be divided into natural aerosols and anthropogenic aerosols, with biological aerosols being a part of natural aerosols.

From a chemical composition perspective, aerosols are classified as organic or inorganic. Based on elements, the bioprotein aerosols was expected to be included in the organics. However, we noticed that when k = 2, all particles can be clustered to either organics or bioprotein aerosols. When k ranges from 2 to 6, about 12% insoluble particles in $M_i$ were clustered consistently as bioprotein aerosols. It indicates that the centroid of bioprotein aerosols differ from those representing organics, primarily due to their nitrogen content. It is crucial to acknowledge this distinguishing feature, leading to designate bioproteins as a single cluster.

• Line 288: Shouldnt the geometric mean diameter be Dg or dg, n ot rg? r could be too easily confused with radius.

Reply:

It is a typo. The "rg" should be "Dg".

• Fig. 9 is much too small to make out necessary detail. While I appreciate the authors conscientiousness in ensuring the reader is aware of the geographical locations where the hailstone samples were sourced, I think the responsibility rests with the reader at this point and the maps are no longer necessary. I would split this figure into 2-3 figures to allow points to be made about each individually, as there is a lot of information here. Plus, more detail can be gleaned.

Reply:

The Fig. 9 has been modified as below:

[Figure]

Fig. 9a. Size distribution of insoluble particles within the natural shells of 12 hailstones is represented. Blue triangles, orange squares, and purple diamonds are used to indicate dust, organics, and bioprotein aerosols, respectively. The natural shells are denoted alphabetically with capital letters (shell A refers to embryos, and shell B/D refers to the crust of hailstones). The arrow direction illustrates the tendency of particle number concentration in each layer compared to the previous shell. Shading is employed to indicate uncertainty. Detailed calculations are provided in the supplementary information. Abbreviations (corresponding to Table. 1): BJ - BeiJing; BS - BaiSe; FS - FuShun; GY - GuYuan; GYA - GuiYAng; YT

[Figure]

Fig. 9b is a continuation of Fig. 9a.

• Section 3.4: Is this section about particle concentrations from the embryos, the shells, or both? Are there concentration size distributions of these particle types for the air at large, or in emissions from specific cities, that these distributions could be compared to? What does having these equations accomplish?

Reply:

The particle concentrations discussed in this section were based on entire hailstones, i.e. both the embryos and the shells.

We have compared particle size distribution with local PM2.5, PM10, total aerosol optical depth, organics concentration and dust concentration in the MERRA2 (Modern-Era Retrospective analysis for Research and Applications, Version 2) dataset. However, we did not find a strong correlation among them. One possible reason is that the data for PM2.5 and PM10 are daily averages. Additionally, the horizontal spatial resolution of the MERRA2 dataset is 0.5° × 0.625°, which may not able to accurately capture local convective processes.

The size distribution of insoluble particles in the entire hailstone records information of insoluble particles within the hailstorms, along the path of hail growth. The size distributions share one thing in common: dust particles in convective clouds exhibit a larger size and a higher particle number concentration compared to organics.

Additionally, when the number concentration (std cm$^{-3}$) of aerosol particles with diameters larger than 0.5 μm ($n_{aero,0.5}$) ranges from 0.1 std cm$^{-3}$ to 10 std cm$^{-3}$, the number concentration (std L$^{-1}$) of ice-nucleating particles ($n_{IN,T_k}$) shows a variation of one to two orders of magnitude (DeMott et al., 2010). The difference in total concentration of particles from different cases is about two orders of magnitude. Therefore, it is crucial to consider the total aerosol number concentrations in the freezing parameterization for a local deep convection simulation.

• Lines 325-326: Im not sure why this statement couldnt be gleaned from Fig. 8 alone, without needing to fit to Eq. 12.
Reply:
The Fig. 8 represents the number concentration of insoluble particles inside hailstone embryos, while Eq. 17 was utilized to fit the particle size distribution that across the entire hailstone. The size distribution of insoluble particles within the entire hailstone record aerosol information within the hail growth region, including particles collected from supercooled water, ice crystals, and air.

**Grammatical/Typographical corrections:**

There are quite a few minor grammatical errors throughout, things like "a" or "the" missing before words, misplaced commas, or subject/verb tense agreement. These don't obscure the science being presented, but I recommend the authors ask for proofreading help from a source with professional proficiency in English. I've included some examples from the first couple pages below.
We will make every effort to polish the text.

• Line 13: "to little regard paid to"
Text has been revised.

• Line 16: "A total of 289,461..."
Text has been revised.

• line 17: comma after bioprotein
Text has been revised.

• Line 17: vary → varies, in → among
Text has been revised.

• Line 18: "were performed as" → "were found to follow"
Text has been revised.

• Throughout: need a space between last letter of a word and the first parenthesis of a citation
Text has been revised.

• Line 27: "that leads" → "leading"
Text has been revised.

• Line 27: Add "the" before "number concentration"
    Text has been revised.

• Line 235: "graupels" → "graupel particles"
    Text has been revised.

**Reference:**

Calinski, T. and Harabasz, J.: A dendrite method for cluster analysis, Commun. Stat. - Theory Methods, 3, 1–27, https://doi.org/10.1080/03610927408827101, 1974.

Davies, D. L. and Bouldin, D. W.: A Cluster Separation Measure, IEEE Trans. Pattern Anal. Mach. Intell., PAMI-1, 224–227, https://doi.org/10.1109/TPAMI.1979.4766909, 1979.

DeMott, P. J., Prenni, A. J., Liu, X., Kreidenweis, S. M., Petters, M. D., Twohy, C. H., Richardson, M. S., Eidhammer, T., and Rogers, D. C.: Predicting global atmospheric ice nuclei distributions and their impacts on climate, Proc. Natl. Acad. Sci., 107, 11217–11222, https://doi.org/10.1073/pnas.0910818107, 2010.

Fan, J., Zhang, Y., Wang, J., Jeong, J., Chen, X., Zhang, S., Lin, Y., Feng, Z., and Adams-Selin, R.: Contrasting Responses of Hailstorms to Anthropogenic Climate Change in Different Synoptic Weather Systems, Earth's Futur., 10, 1-20, https://doi.org/10.1029/2022EF002768, 2022.

Kohonen, T.: The self-organizing map, Proc. IEEE, 78, 1464–1480, https://doi.org/10.1109/5.58325, 1990.

Lamb, D. and Verlinde, J.: The atmospheric setting, in: Physics and Chemistry of Clouds, Cambridge University Press, 29–122, https://doi.org/10.1017/CBO9780511976377.003, 2011.

Li, X., Zhang, Q., Zhu, T., Li, Z., Lin, J., and Zou, T.: Water-soluble ions in hailstones in northern and southwestern China, Sci. Bull., 63, 1177–1179, https://doi.org/10.1016/j.scib.2018.07.021, 2018.

Rousseeuw, P. J.: Silhouettes: A graphical aid to the interpretation and validation of cluster analysis, J. Comput. Appl. Math., 20, 53–65, https://doi.org/10.1016/0377-0427(87)90125-7, 1987.

Sibson, R. and Hartigan, J. A.: Clustering Algorithms., Appl. Stat., 25, 70, https://doi.org/10.2307/2346526, 1976.

Taylor, J. R.: An Introduction to Error Analysis, Second edi., University Science Books, 330 pp., ISBN 093570275X, 1997.

---

## Author Response (AR2)

**egusphere-2023-290**

**Reply to editors:**

Based on review reports from three referees, I'm happy to accept your paper published in ACP if the following two minor points could be addressed.

Thank you for your kind comments. We have made effort to clarify every comment and implemented all the suggestions in the revised manuscript. The following texts are our point-to-point response.

1) About the last sentence in the abstract "Our finding suggests the aerosol species and number concentration variance in different storms should be considered in model simulation of the ice freezing process", this statement is somewhat excessive. It looks impossible to include the number concentration variance in the model. Please delete it or rephrase it as "Our findings highlight the need for atmospheric chemistry to be considered in the simulation of ice freezing process".

We have rephrased the sentence as your advice.

2) Fig. 1 and Fig. 8: Please show the full map in Fig. 1 and remote the background map in Fig. 8. The origin Fig.1 and Fig. 8:

In Fig. 1, we present the full map, while in Fig. 8, we have excluded the background. The caption for Fig. 1 remains unchanged. Below, you can view the two updated figures:

[Figure]

**Fig. 1: Geographical distribution of collected hailstones. The collecting locations of hailstones are indicated by black dots. Provinces of China from which the hailstones were collected are represented by six different**

colors. The number of hailstones we analyzed was indicated in parentheses. Abbreviations (corresponding to Table 1): BJ - BeiJing; BS - BaiSe; FS - FuShun; GY - GuYuan; GYA - GuiYAng; YT - YanTai.

[Figure]

Fig. 8: Size distribution of (a) organics, (b) dust, and (c) bioprotein aerosols in hailstone embryos. Colors represent different hailstones. Abbreviations (corresponding to Table 1): BJ - BeiJing; BS - BaiSe; FS - FuShun; GY - GuYuan; GYA - GuiYAng; YT - YanTai.

3) Fig. 10: Some texts are overlapped by the figure plots. Please improve these plots, or maybe consider adding a table (as SI) to show all the equations and parameters.

Following your recommendation, we have excluded the parameters pertaining to image coverage and encapsulated them in Table 2.

[Figure]

**Fig. 10: Fitting size distribution functions of organics and dust contained in the whole hailstone. (a)-(h) Fitting parameters of logarithmic normal distributions of BJ1, BJ2, BS, FS, GY1, GY2, YT, GYA. (i) Classic modes of dust and organics (interval of data is 0.2 μm and fitting curves painted with interval of 0.02 μm). The fitting parameters for subfigures (a)-(h) are listed in Table 2. The fitting range of (a)-(h) is shown with a green rectangle in (i). The centroid of the organics fitting parameter (orange line) is $\ln \sigma_o = 0.91$, $\ln D_o = -0.70$, and $N_o = 9.19 \times 10^5$ cm$^{-3}$. The centroid of the dust fitting parameter (blue line) is $\ln \sigma_d = 1.07$, $\ln D_d = 0.11$, and $N_d = 1.59 \times 10^6$ cm$^{-3}$. Shading showed uncertainty of organics and dust. Abbreviations (corresponding to Table 1): BJ - BeiJing; BS - BaiSe; FS - FuShun; GY - GuYuan; GYA - GuiYang; YT - YanTai.**

**Table 2:The fitting parameters of dust and organics size distribution in Fig. 10 (a)-(h).**

| Sample | $N_o$ $(cm^{-3})$ | $ln\,D_o$ | $\ln\sigma_o$ | $R_o^2$ | $N_d$ $(cm^{-3})$ | $ln\,D_d$ | $\ln\sigma_d$ | $R_d^2$ |
|--------|--------|--------|--------|--------|--------|--------|--------|--------|
| BJ1 | $4.57\times10^5$ | -0.98 | 0.90 | 0.97 | $7.11\times10^5$ | 0.20 | 1.06 | 0.93 |
| BJ2 | $9.32\times10^4$ | -0.90 | 0.88 | 0.98 | $2.55\times10^5$ | 0.02 | 1.01 | 0.89 |
| BS | $6.65\times10^5$ | -0.75 | 0.98 | 0.97 | $4.12\times10^5$ | 0.40 | 0.84 | 0.91 |
| FS | $4.13\times10^5$ | -1.12 | 0.93 | 0.89 | $2.35\times10^5$ | -0.05 | 1.15 | 0.87 |
| GY1 | $2.66\times10^6$ | -0.05 | 0.69 | 0.97 | $8.15\times10^6$ | 0.57 | 0.96 | 0.98 |
| GY2 | $1.60\times10^6$ | 0.10 | 0.79 | 0.98 | $1.25\times10^6$ | 0.37 | 1.06 | 0.95 |
| YT | $1.21\times10^6$ | -0.90 | 0.87 | 0.98 | $1.16\times10^6$ | 0.20 | 0.92 | 0.94 |
| GYA | $2.51\times10^5$ | -0.99 | 1.21 | 0.84 | $5.06\times10^5$ | -0.87 | 1.57 | 0.79 |

4) The font sizes of all figures/sub-figures could be somehow improved.

We have meticulously revised all figures and sub-figures, striving for full satisfaction.

5) Figure 9: Please note that single figure panels with their own caption are not allowed in the final version.

The caption has been improved.

**Author's annotation in other one modification:**

1) We found an error in Fig. 5 and corrected it.

---

## Author Response (AR3)

**egusphere-2023-290**

**Reply to editors:**

Please address the below two technical points that raised by the editorial office.

Thank you for your thoughtful feedback on our manuscript. We appreciate your positive comments and would like to express our gratitude for your time and effort in reviewing our work. Below, we provide a point-to-point response to your valuable comments:

1. Figure 9: Please note that single figure panels with their own caption are not allowed in the final version.

We placed the caption under all the panels of this figure.

2. It seems that table is included as figure #6. If it is so, it must be re-labelled as table and the references in the manuscript text must be adjusted accordingly.

We relabeled figure #6 as table #2 and the references in the manuscript text have been appropriately updated to align with this change.